# A Model for Cost–Benefit Analysis of Privately Owned Vehicle-to-Grid Solutions

**Jesús Rodríguez-Molina** [1,*] **, Pedro Castillejo** [1] **, Victoria Beltran** [2] **and Margarita Martínez-Núñez** [3]

[1] Department of Telematics and Electronics Engineering, Technical University of Madrid, 28031 Madrid, Spain; pedro.castillejo@upm.es

[2] Department of Electronics, Computer Technology and Projects, Technical University of Cartagena, 30203 Cartagena, Spain; victoria.beltran@upct.es

[3] Department of Organization Engineering, Business Administration and Statistics, Technical University of Madrid, 28031 Madrid, Spain; margarita.martinez@upm.es

\* Correspondence: jesus.rodriguezm@upm.es; Tel.: +34–910673350

**Abstract:** Although the increasing adoption of electric vehicles (EVs) is overall positive for the environment and for the sustainable use of resources, the extra effort that requires purchasing an EV when compared to an equivalent internal combustion engine (ICE) competitor make them less appealing from an economical point of view. In addition to that, there are other challenges in EVs (autonomy, battery, recharge time, etc.) that are non-existent in ICE vehicles. Nevertheless, the possibility of providing electricity to the power grid via vehicle-to-grid technology (V2G), along with lower maintenance costs, could prove that EVs are the most economically efficient option in the long run. Indeed, enabling V2G would make EVs capable of saving some costs for their vehicle owners, thus making them a better long-term mobility choice that could trigger deep changes in habits of vehicle owners. This paper describes a cost–benefit analysis of how consumers can make use of V2G solutions, in a way that they can use their vehicle for transport purposes and obtain revenues when injecting energy into the power grid.

**Keywords:** electric vehicle; vehicle-to-grid; cost–benefit analysis

## 1. Introduction

The smart grid is one of the most promising infrastructures developed during the last years for the improvement of access to electricity and its usage, as it is bringing key benefits: a combination of existing information and communication technology standards [1], the power grid itself to enhance the stability of the system [2,3], and the incorporation of new actors in the energy markets [4]. Among other features, the smart grid enables a set of activities aimed to the demand side management, used to optimize energy usage according to specific characteristics of demand response systems, energy efficiency, or usage time of the resources [5,6]. Other applications such as home load control and home energy management [7] are covered as well. Energy storage is a major feature, due to the fact that it has to be enabled and balanced in distributed-like systems for increased effectiveness [8] and can be used to trade it in the aforementioned energy markets or to provide energy in moments where it cannot be harvested from the environment (like photovoltaic deployments during the night). More importantly, it allows prosumers (that is, energy consumers able to produce their own electricity by means of distributed energy resources) to have more energy available for their private use and utilize the surplus power they produce as a source of revenues. In this regard, electric vehicles (EVs) may become an appealing solution—especially when compared to vehicles with an internal combustion engine (ICE)—as they are capable of having vehicle-to-grid (V2G) characteristics that will enable them to inject electricity into the

power grid, resulting in an opportunity to create income for the V2G vehicle's owner. To this end, it is necessary to add some specific infrastructure to the EV, namely, a V2G bidirectional power converter, or to change the software configuration of its electric charger. Furthermore, this V2G approach leads to new trade opportunities that were not possible before, as for example selling energy to an aggregator located between the distributed system operator and the prosumers.

This paper studies how privately owned V2Gs can compete with ICE-based vehicles in terms of economic efficiency, putting forward some scenarios where a new mathematical model has been demonstrated. The cost–benefit model that is presented in this manuscript shows a thorough comparison of the expenses between EVs and ICE vehicles during an extended period of time, as well as an economic assessment between purchasing and renting the battery of an EV and how costs vary depending on several different profiles of vehicle usage. The authors have established a comparison between ICE and V2G vehicles because the objective of the manuscript is to assess if V2G technology can be used to make EVs economically competitive when compared to the traditional ICE-powered automobiles. Typically, and especially if subsidies are removed, the cost of an EV is higher than a comparable ICE vehicle. Even though maintenance costs and electricity are lower than gas and maintenance of ICE-powered vehicles, it is at least arguable whether at the end of its timespan of usage an EV is more economical than an ICE vehicle. Nevertheless, by using V2G technology, an EV should be better positioned to reduce costs in mobility with privately owned vehicles. It is the authors' opinion that it is interesting to have a study on the matter of comparing ICE vehicles with EV-V2G ones, as it could provide a more accurate perspective on how advantageous it is to use V2G technology in an EV to reduce expenses. Considering a set of parameters and how they relate with each other, an analysis of the obtained calculations has been carried out for certain cases. The contributions of this paper are as follows:

1.  A thorough mathematical cost–benefit model used to analyse accurately to what extent V2G technology can be profitable for a regular end user. This model takes into account parameters that, to the best of the authors' knowledge, have not been included in any other cost–benefit model that involves different end users for V2G technology depending on their vehicle user profile, like battery discharging while being idle, how the depreciation of the electric vehicle influences its maintenance costs, or the suboptimal trade of electricity that might happen if the vehicle is not available during the most suitable time slots of the day to be recharged.
2.  The differences in expenditures for V2G solutions when the battery is purchased with the whole vehicle or leased from the manufacturer. This is another contribution that the authors of this paper have not seen in the existing literature about this topic.
3.  The application of the cost–benefit model to three user profiles for V2G and ICE solutions, along with how they fare after a prolonged period of time. The authors believe that this adds a realistic justification with several examples that make use of actual data and parameter values extracted from updated references in order to know to what extent using V2G may provide an economical benefit to their end users.
4.  A review of comparable models that have been created by other authors, pointing out the main challenges that have still to be dealt with and why the one put forward by the authors of this manuscript represents an improvement over the previous ones.

This paper is structured as follows: an introduction has already been offered as the first section. Section 2 offers a compilation of related works. Section 3 describes the variables included to elaborate the model. Section 4 presents the model. Section 5 offers the numerical evaluation of the model when facing two different possible scenarios. Section 6 explains the conclusions obtained from the study. Acknowledgments and references are displayed as the final parts of the manuscript.

## 2. Related Works

The studies done about the possible applicability of a V2G solution for particular environments have been included in this section, along with the open issues that have been found in the reviewed literature.

### 2.1. State of The Art

L. Noel and R. McCormack put forward their own cost–benefit analysis when comparing a V2G-capable electric school bus with a diesel-powered one [9]. They take into account a large set of variables (including seating capacity, cost of electricity, cost of diesel fuel, etc.), the authors conclude that using a school bus with V2G capabilities is more cost-effective than a diesel one when V2G capabilities are enabled, thus making the latter almost mandatory (savings up to $6070 per seat are claimed). However, the study focuses on municipal school buses (which are more expensive and far less abundant than automobiles), rather than private transportation. This study has been challenged by the one presented by Y. Shirazi et al. [10], where it is mentioned that, as far as Philadelphia and its school district are concerned, a V2G bus is not cost effective and it actually increases its usage costs compared to a diesel-powered one. The reasons behind this conclusion involve limitations that, according to the authors, are inherent to electric vehicles and are often overlooked, such as low environmental temperatures or electrical losses resulting from V2G technology.

D. Park et al. offer a cost–benefit analysis where it is claimed that savings with EV services range from $8000 to $22,000 per year and per vehicle in an optimized frequency regulation (FR) market [11], which is the one that best adapts to the nature of V2G services, due to its pattern of energy supply in bursts rather than as a constant and reliable flow source. The authors consider fine-grained characteristics like daily mobility patterns and mobility model velocities. The study that has been done here, though, only covers municipal services (school transport, waste collecting truck, and city bus) rather than private vehicles.

O. A. Nworgu et al. describe the economic prospects of V2G technology in the electric distribution network [12]. They mention how V2G infrastructure can be used for valley filling during low demand periods and peak shaving when electricity demand is high. However, their model does not take into account the energy losses resulting from using V2G as a way to store and transfer energy (rather than a regular generator or home battery) or the required cost to adapt an EV to V2G technology.

D. M. Hill et al. describe fleet operator risks for V2G regulation [13]. A V2G fleet financial model is displayed where the replacement of ICE trucks with extended range electric vehicles is studied, considering three scenarios where this replacement may or may not be cost efficient. Battery degradation and replacement, which easily comes as one of the most significant challenges of V2G technology, are fully considered, as well as risk acceptance for vehicle owners that might be unwilling to switch to this kind of technology. The authors´ proposal, though, is focused on fleets of vehicles rather than private transport.

M. Musio et al. consider the added benefits of having V2G technology working as a virtual power plant (VPP) [14]. The authors stress the importance of having a suitable battery available for this kind of technology and offer a thorough study on a simulation of a battery lifetime in terms of charge and discharge. In addition to that, a case study is displayed where an optimization problem, understood as the number of EVs that minimizes the cost of the VPP, is reasoned. However, the authors explicitly mention that the resulting VPP works autonomously with no trade activities with the main grid, as it has likewise been considered in this manuscript.

P. Jain et al. also mention a similar idea with aggregated EVs included in a V2G-based power service [15]. Different kinds of vehicles are taken into account for the estimations done regarding revenue evaluation. The aggregated electricity provided by the V2G network is assessed as the aggregated state of charge (SOC) of the batteries. However, the work presented by the authors deals with specific information that has been obtained from external sources and they perform the calculations based on them, rather than attempting to offer a new model.

H. Lund and W. Kempton describe in [16] how renewable energies can be integrated in the transport sector via V2G. The authors present a model, referred to as EnergyPLAN, which they have developed under a framework of national level energy devoted for transport, heat and electricity. V2G plays a prominent role in this model, due to the fact that the sharing of vehicles enabled with this technology that is connected to the grid is expected to provide power to the grid. The number of

inputs that have been used in the model to define EVs with V2G are fewer than the ones that have been considered in this manuscript, though. The authors have considered the transportation demand of electric cars, share of V2G solutions both being driven during peak hours and connected to the grid, efficiency of the chargers and inverters, capacity of the battery, distribution of the transportation demand, and the power capacity of the grid connection.

H. Qiang et al. put forward a mathematical model [17] where the initial SOC, charging power and initial charging time are assessed with the objective of obtaining a more accurate way to compute the charging load used by private EVs. Their model takes into account the SOC of the battery, the initial SOC of charging and the charging power, but it falls short when considering other features more related to an economical point of view, such as battery degradation, inflation or the battery costs.

Santoshkumar et al. propose an architectural framework of an off-board V2G integrator for the Smart Grid [18]. They refer to off-board integrators as the ones that are outside of vehicles and are able to connect several EVs to the power grid. In the mathematical model that they put forward there are several features that have been taken into account for the testing activities that the authors have carried out: power of the domestic loads efficiency of the chargers or the number of existent EVs are some of them. Unfortunately, the features involved by the scope of this manuscript, which are used to demonstrate the economic feasibility of the integration of V2G technology in the smart grid are not present.

Chenggang Du and Jinghan He also mention how a strategy for multiple V2G solutions can be applied for their batteries' charge and discharge [19]. According to the authors, this charge–discharge plan would be able to lower differences between peak and valley energy demand hours significantly. Among other characteristics, power and energy restrains are taken into account to create the daily load curve that is obtained after enhancing daily energy consumption with the integration of V2G technology. As it happened with some previous proposals, this one models quite accurately features related to electricity and power but does not take into account the potential economic benefits of V2G owners.

Zesen Wang et al. describe in [20] their own contributions to the usage of V2G technology for building-integrated energy systems (referred to as BIES). They determine how vehicles with this technology can be used as movable energy storage devices capable of providing electricity to other loads. V2G plays a supportive role in the suggested model, as simulations have been used to prove that a fleet of V2G equipment can improve the overall economy of BIES. However, the authors of this paper have focused on the role of V2G within a BIES, rather than making a BIES part of the grid or focusing it as a specific solution for end users.

Yuancheng Li et al. show in [21] how differential privacy is an important matter to consider in V2G networks. The overall structure of a V2G is introduced, and the roles of each of its entities (control center, aggregator, distribution network, and charge station) are described as well. As far as privacy protection is concerned, a spatial data decomposition algorithm is put forward by the authors. Experimental results obtained from the charging positions of 100,000 electric vehicles and 1500 public charging posts have been presented. However, the researchers´ main purpose is to address differential privacy in the charging infrastructure of V2G networks, rather than presenting a cost–benefit analysis.

Tohid Harighi et al. make an overview of storage systems, energy scenarios and the required infrastructure for V2G technology [22]. It is regarded as part of the overall infrastructure that would be required to decrease greenhouse gases (GHG) to an acceptable minimum that meets the targets that have been agreed for 2050. Unfortunately, the paper does not offer a mathematical model on how to integrate V2G technology in a larger network, nor it provides a cost benefit analysis on the profit possibilities offered by V2G.

Michael Child et al. estimate in [23] how a significant amount of V2G solutions could impact a completely renewable system. The authors of this manuscript have used the above-mentioned EnergyPLAN modelling tool as a way to assess the impact of the contributions that can be done by a V2G network. A thorough assessment on how energy would be consumed, supplied and stored is made in the manuscript. There is no cost–benefit analysis model presented by the authors, though.

Another study based on comparisons between long-term usage of EVs and ICE vehicles is the one made by Peter Weldon et al. in [24]. The authors show how, under the specific use case of Irish infrastructure and economic incentives to buy EVs, different levels of economic competitiveness of EVs over ICE vehicles can be achieved. The authors have studied four different kinds of comparable vehicles (small, medium, large, and vans) for both kinds of energy sources and have reached several conclusions: after a 10-year period of time, EVs are more economically efficient in almost every possible situation, except when gasoline prices remain constant. Overall, the paper describes the situation that would take place in scenarios where vehicles have high, medium, or low frequency of usage and the conclusions reached are close to the ones that we have obtained as well. However, battery degradation is not considered as detailed as in this manuscript, nor there is information on efficiency with V2G solutions. Lastly, externalities are not taken into account, and battery replacement is only considered for the high frequency usage case, which is to be expected since regular EVs that do not make use of V2G facilities should not require such an action.

A similar study is shown by Yiling Zhang et al. [25]. In this case, V2G has been studied as a technology oriented to car sharing. In order to quantify the potential benefits from using it, a model making use of two-stage stochastic integer program has been considered. An estimation of the benefits of integration has been made by the authors, which includes the benefits that will result from the energy trade, as well as costs related to vehicle relocation and charging. This study, though, is not targeting battery degradation as a major factor as it is done in our manuscript, and there are no different user profiles for the model that has been created.

In the piece of research made by Kyuho Maeng et al. [26] the integration of V2G into the grid and what benefits it can provide are major topics for research as well. The authors of this paper offer a mixed multiple discrete-continuous extreme value (MDCEV) model based on random utility theory (RUM). The model is used to obtain market simulation results that define what kind of vehicle would be preferable for a sample of Korean population. This study, though, does not consider profitability for end users as one core concept, nor battery degradation is taken into account in a thorough manner.

There are also other references that consider externalities for V2G technology. For example, it is shown in [27] that, "BE [Battery Electric] transit and school buses with V2G application have potential to reduce electricity generation related greenhouse-gas emissions by 1067 and 1420 tons of CO2 equivalence (average), and eliminate $13,000 and $18,300 air pollution externalities (average), respectively". Air externalities are compared between V2G and ICE (diesel) mobility solutions, along with the V2G technology cost for similar school and transit buses. According to this manuscript, in the CAISO (California ISO) region, V2G makes possible that the lifetime total cost of an electric school bus is little more than a sixth of the cost in the diesel one, whereas costs for a regular transit bus are around a fifth lower for V2G than for the ICE solution. However, as it happened in other cases, the study is not applied to private transport. In addition to that, it is stated in [28] that if externalities are taken into account for generation, new storage (where V2G solutions are included) and new loads to model a large regional transmission organization, 50% of renewable energy should be implemented. This study is more focused on externalities than in V2G usage, though. The usage of V2G combined with other smart grid technologies has been subject of research as well. For example, demand response (DR) is the main focus on [29]. The study proves how using V2G in specific moments such as night time can improve the overall regularity of energy consumption (a feature most looked into from the point of view of the electricity supplier) with the aid of a home energy management (HEM) system, smart meters and V2G itself. It is also mentioned that V2G can put a strain on loads working during the night, as they can increase in number due to low energy prices during that time slot. The interaction between demand response management and V2G is also studied in [30], where it is explained that their cooperation is critical to use surplus energy in EVs to the end user´s advantage. The system that is put forward takes an auction-like approach: by means of having EVs selling electricity under dynamic pricing to a number of aggregators, the latter compete to obtain the best possible price, while at the same time offering incentives to EVs to act as V2G solutions.

Finally, there are some more studies that have researched on the economic and energy charging possibilities of comparable EV and EVV2G solutions. For example, it is claimed in [31] that dynamic EV scheduling charge/discharge can optimize V2G usage and capacity. The authors of the manuscript describe how an algorithm built as part of their building energy management system (BEMS) can be used for 30 min V2G capacity estimations. Their model has been tested for three different use cases (high-rise residential buildings, office buildings, and commercial buildings) and the researchers mention how using several EVs as distributed energy storage can be possible for high-rise buildings. Long-term costs compared with vehicles with ICE-powered vehicles is out of the scope of the manuscript, though. Additionally, it is studied in [32] how different charging schemes with or without the usage of V2G can offer complementary results. The authors discuss four charging modes (night charging, night charging with V2G, 24 h charging, and 24 h charging with V2G) and study how they impact in vehicle usage. It is also mentioned how V2G provides an opportunity to profit through electricity arbitrage by discharging energy to the power grid during non-driving periods of time. This piece of work, however, is focused on the different charging possibilities for an EV rather than its long-term economic performance compared to the one that an ICE vehicle offers. Lastly, a model for communications based on the long term evolution (LTE) protocol among EVs that make use of V2G technology is described in [33]. The researchers claim how this protocol can be used to communicate two EVs wirelessly by making use of the physical layer present in the LTE protocol. In this way, it is claimed that an aggregator can send information to EVs about power requirements on an area under its range, and in case a V2G is unaware of the power demand, the LTE system will send the information from a regular EV to a V2G automobile. State of charge in the battery of the EV is the main feature used to establish whether power will be bought or sold.

The most prominent features from the reviewed literature have been included in Nomenclature. Many of these studies' strong points have been taken into consideration for the mathematical model that is presented in this manuscript. For example, battery degradation and replacement are a major part of the studio that has been done, whereas weaknesses in Table 1 like lack of attention to private transport have been sufficiently covered in the mathematical model put forward in this manuscript.

**Table 1.** Summarization of the main advantages and disadvantages of the reviewed literature.

| Reviewed Work | Strengths | Weaknesses |
|---|---|---|
| L. Noel and R. McCormack [9] | Complete cost–benefit analysis for public transport | Focused on school buses rather than private transportation. This manuscript has been challenged by [10] |
| Y. Shirazi et al. [10] | Provides more parameters to consider (low temperatures, electrical losses) | Focused on school buses rather than private transportation |
| D. Park et al. [11] | Mobility patterns and mobility model velocities are taken into account | The study only covers municipal services |
| O. A. Nworgu et al. [12] | It is mentioned how to use V2G to flatten demand curve | The model does not take into account energy losses from using V2G |
| D. M. Hill et al. [13] | Battery degradation, replacement and risk acceptance are taken into account | The proposal deals with fleets of vehicles rather than private transport |
| M. Musio et al. [14] | The optimization problem resulting from having vehicles acting as a VPP is analyzed | The resulting VPP has no trade activities with the main grid |
| P. Jain et al. [15] | Perspective on SOC of the batteries is provided | Calculations are done based on external sources rather than by providing a new model |
| H. Lund and W. Kempton [16] | Model that integrates energy used for transport, heat and electricity at a national level | Less variables are taken into account than in the model presented in the manuscript |

**Table 1.** *Cont.*

| Reviewed Work | Strengths | Weaknesses |
| --- | --- | --- |
| H. Qiang et al. [17] | Initial SOC, charging power and initial charging time are considered | Battery degradation, inflation or the battery costs are not considered |
| Santoshkumar et al. [18] | Varied loads have been taken into account in the model | economic feasibility of V2G integration is not present |
| Chenggang Du and Jinghan He [19] | Power and energy restrains are used for the daily load curve | The model does not take into account V2G owners |
| Zesen Wang et al. [20] | V2G solutions are modelled as movable energy storage devices | The model regards V2G as a support for Building-Integrated Energy Systems |
| Yuancheng Li et al. [21] | A thorough experimental analysis has been done regarding location privacy | The model focuses on differential privacy in V2G rather than doing a cost–benefit analysis |
| Tohid Harighi et al. [22] | V2G is acknowledged as a technology that can be used to meet goals in GHG reduction | Neither mathematical model nor cost–benefit analysis for V2G are offered |
| Michael Child et al. [23] | Impact of V2G in a system completely based on renewable energies is assessed | No cost benefit analysis has been performed in the manuscript |
| Peter Weldon et al. [24] | Model with three different kinds of EV users | No data about V2G solutions or battery degradation. Externalities not considered |
| Yiling Zhang et al. [25] | Study on the integration of V2G into the electricity grid | No data about battery degradation. No different profiles |
| Kyuho Maeng et al. [26] | Study of the most preferable kind of vehicle for a significant sample of users | No data about battery degradation. Profitability for end users is not considered |

*2.2. Open Issues*

There are several open issues that required to be tackled if an accurate, objective assessment of V2G technology is going to be done.

1.  Limitations in the mathematical models. Despite the efforts done by the authors, the mathematical models used show limitations that make them obsolete after a relatively short amount of time. A model that offers a significant number of parameters to measure accurately the cost–benefit of V2G solutions, while at the same time keeping the time required to perform calculations at a reasonable level, is required.

2.  Lack of orientation to private transport. The evaluations that are done in the literature are mostly concerned about transport fleet or public services. However, the adoption of these solutions by private owners of vehicles is a critical point for V2G, as they are more numerous and the cost of their means of transport is lower when compared to a school bus or a truck. While there are literature references proving that EV owners could be interested in enabling their vehicles with V2G technology if given suitable options ("Our findings suggest that the V2G concept is most likely to help EVs on the market if power aggregators operate either on pay-as-you-go basis", [34]) they do not show a mathematical model that takes into account different EV user profiles, externalities or battery usage options.

3.  Reduced scope of numerical results. The studied literature usually reflects how a model can be applied or not by taking into account too specific situations, such as public transport in a city or any other location that is very dependent on meteorological circumstances, the kind of public service that is attended or the route that is taken every day by the vehicles, so it becomes difficult to make an objective analysis of those scenarios.

All these challenges have been born in mind to design the mathematical model presented in this manuscript, as well as the calculations and results placed in the next sections. Overall, the model can be described as depicted in Figure 1. The main figures that have been taken into account are acquisition

and operational costs, externalities inherent to the vehicle, gas consumption, and maintenance, among others.

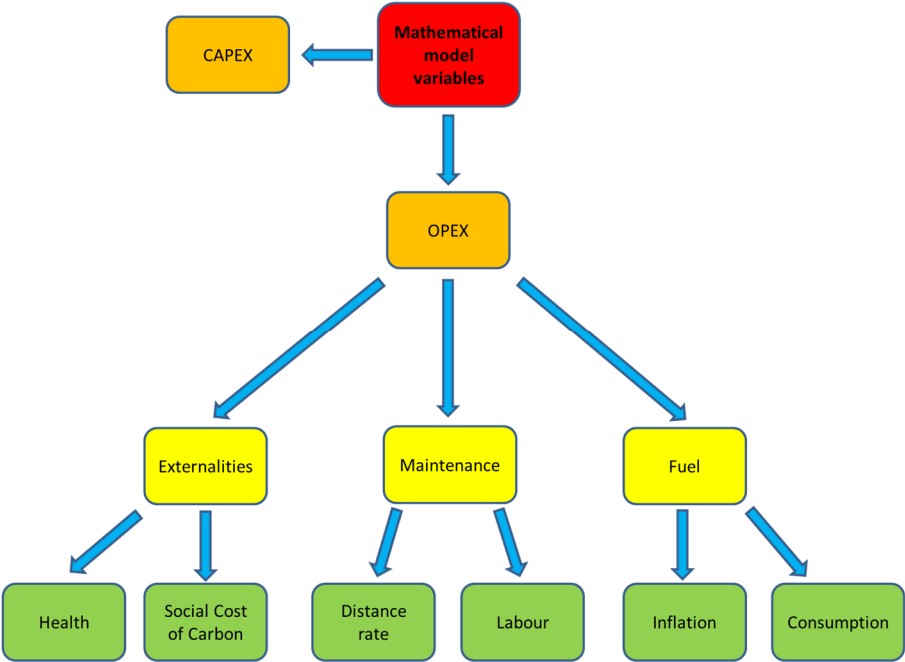

**Figure 1.** Common variables considered for the mathematical model and their relations.

## 3. Mathematical Model for V2G Integration

The costs that have to be faced by an individual (or a small group like a family that will use a single automobile) can be defined as $C_{totICE}$ for the ICE vehicle, and $C_{totV2G}$ for the vehicle-to-grid solution, whereas *Capex* figures for the ICE and the V2G solution could represent the cost of acquiring the vehicle by an individual. Finally, *Opex* figures for the ICE and the V2G vehicles represent the mandatory operational expenses needed to have the asset fully functional. Thus, the total costs for each of the transport solutions can be expressed as in (1) and (2):

$$C_{totICE} = Capex_{ICE} + Opex_{ICE} \tag{1}$$

$$C_{totV2G} = Capex_{V2G} + Opex_{V2G} \tag{2}$$

Note that it has been chosen to consider each of the vehicles as an asset rather than a liability due to V2G potential to generate revenues or at least reduce operational costs. Inflation can also be considered in the *Opex* expenditures by adding its corresponding parameter (represented by *Inf*) into the previous equations, as long as it is defined for a specific amount of time. Typically, inflation will build up as time goes by as a function, or a part of one, where time piles up on an exponential basis. Therefore, inflation-adjusted prices have been added as shown in (3) and (4).

$$C_{totICE} = Capex_{ICE} + \sum_{i=1}^{j} (1 + Inf)^i \times Opex_{ICE} \tag{3}$$

$$C_{totV2G} = Capex_{V2G} + \sum_{i=1}^{j} (1 + Inf)^i \times Opex_{V2G} \tag{4}$$

Additionally, *Capex* for the EV must be further defined as the costs of installing the required infrastructure to transform the EV into a vehicle with V2G capabilities (represented as *Vconv*) and to

charge the EV at home (represented by *Heq*), along with the cost of the vehicle itself ($C_{EV}$). It has been done in (5). As it can be inferred, these technological needs do not apply for the internal combustion engine vehicle.

$$Capex_{V2G} = C_{EV} + Vconv + Heq \tag{5}$$

The *Opex* for ICE and EV must be further analyzed. It is shown in (6) what the *Opex* value is for an ICE vehicle. During a period of time that ranges from *i* to *j*—considering *i* as a year and *j* as twelve, which has been estimated as the average lifetime of a vehicle, according to a) what is used in [24] and b) the estimation done in [25]—four aspects will add up to the final figure of the expenses: the yearly costs of the externalities of the vehicle ($Ex_{ICE}$), fuel consumption ($F_{ICE}$), and maintenance expenses ($M_{ICE}$). As far as the V2G solution is concerned the equation is represented by the same kind of terms used for the ICE vehicle (7). Nevertheless, contrary to ICE vehicles, in this case, the expenses, maintenance and electricity costs need to be further defined, as it is detailed through the following subsections.

$$Opex_{ICE} = \sum_{i=1}^{j} (Ex_{ICE} + M_{ICE} + F_{ICE})_i \tag{6}$$

$$Opex_{ICE} = \sum_{i=1}^{j} (Ex_{ICE} + M_{ICE} + F_{ICE})_i \tag{7}$$

### 3.1. Cost of the Externalities of the Vehicle

The externalities that have been presented will offer different values depending on whether an ICE vehicle or a V2G is used. In the first case, as represented in (8), these externalities will be closely linked to the cost of the health impact caused by ICE-based vehicles ($h_{ICE}$), the distance run with the vehicle (*D*), as well as the average consumption of gas ($Av_{consICE}$), carbon emissions (depicted as $C_{ICE}$ for the ICE vehicle) and the social cost of carbon (*SCC*) during a certain period of time. These externalities are also reflected for the V2G solution in (9), where the equivalent data has been included. Health impact ($h_{V2G}$) and carbon emissions ($C_{V2G}$) are harder to measure in the case of V2G, as they are related to the energy mix from which electricity is coming and, more specifically, the amount of renewable energies present in this energy mix. The cost of the electricity used to move the vehicle ($E_{cons}$) can be determined by the trading operations that can be done by the owner of a V2G automobile (even though it will be usually lower that the cost of oil-based fuels). Overall, the cost of these externalities for society has been represented in a manner resembling the one used in [9], as there were concepts such as SCC or distance that had to be taken into account in the same way as it was done in this related work.

$$Ex_{ICE} = \sum_{i=1}^{j} (h_{ICE} \times D + C_{ICE} \times SCC)_i \tag{8}$$

$$Ex_{V2G} = \sum_{i=1}^{j} (h_{v2g} \times D + C_{V2G} \times SCC)_i \tag{9}$$

### 3.2. Cost of Yearly Fuel Consumption

There are several aspects that must be taken into account when including yearly fuel consumption in the mathematical model. For instance, the cost of the energy bought and the price set to sell it back to the market, so that the end user will use arbitrage to their advantage. This feature will be dependent on, among other aspects, two main factors: a) buying and selling actions that take place during energy cost peak or valley hours (while overall an average price for electricity may differ during the day depending on the user tariff, the V2G infrastructure will take advantage of a peak/valley hours setting, as depicted in [35]), and b) the possibility for the end user of the V2G to buy and sell energy at a suitable time

for their interests. The latter implies that due to the usage of the vehicle or their end users' working schedule, they may not be able to charge completely their V2G during valley hours and sell all the electricity during peak hours. All these factors have been taken into account in the mathematical model presented in this manuscript: while energy is bought and sold in the average prices set for valley hours ($Avc_{buy}$) and peak hours ($Avc_{sell}$), real price of electricity when both purchased and sold is obtained as the combination of $Avc_{buy}$,$Avc_{sell}$ and the addition of four different factors that range from 0 to 1, which effectively describes the percentage of the energy that can be bought and sold during each of the two possible time periods (valley or peak hours). They are used to represent the fact that it will be very difficult for regular end users to buy and sell electricity during all-optimal time periods, so there will be just a majority of power bought and sold when it is best for the end user. They are called $f_{bb}$ (for *factor* of energy *bought* during optimal *buying* period),$f_{bs}$ (for *factor* of energy *bought* during optimal *selling* period),$f_{sb}$ (for *factor* of energy *sold* during optimal *buying* period), and $f_{ss}$ (for *factor* of energy *sold* during optimal *selling* period). The efficiency to buy and sell electricity at the suitable moment has been estimated at 80% (hence, the 0.8 value of $f_{bb}$ and $f_{ss}$), so some power will have to be transferred when it is least optimal for them (estimated at 20%, hence the 0.2 value of $f_{bs}$ and $f_{sb}$). This has been done so because there are some examples in literature that show how a portion of the EV charge is done in suboptimal periods of time. For example, it is shown [36] that there is some charging done halfway through the day, which is usually the daily time period when electricity prices gone from valley to peak in two-levelled tariffs. Equally, it is shown in [37] how charge estimations done can take place around 6 p.m., a time of the day that is often part of peak hours. These principles have been included in Equations (10) and (11), which represent the final cost of buying ($C_{rpbuy}$) and selling energy ($C_{rpsell}$) when suboptimal intervals are included. These equations are defined like this because it is assumed that there are basically two levels of prices with small fluctuations inside them (as seen in [32]).

$$C_{rpbuy} = Avc_{buy} \times f_{bb} + C_{sell} \times f_{bs} \tag{10}$$

$$C_{rpbuy} = Avc_{buy} \times f_{bb} + C_{sell} \times f_{bs} \tag{11}$$

Fuel costs are modelled differently depending on the vehicle that is used as a private transport mean. If the ICE-based solution is used, gas costs will be as shown in (12). It is basically the same way that diesel fuel costs are described in [9] (average fuel consumption $Av_{cons}$ and gas price $Cf_{ICE}$ have been used as variables), with the exception that figures used in this case correspond to private automobiles. The equation in (13) shows how yearly costs would be for the V2G solution. Unlike an ICE automobile, it relies heavily on the trading activities that are done with the energy stored in the battery of the V2G vehicle, which imply buying and selling energy (represented in the formula as $E_{buy}$ and $E_{sell}$) to different costs: one to buy it—$C_{rpbuy}$—and a different one to sell it—$C_{rpsell}$. Buying prices are expected to be lower than selling ones; otherwise, the opportunity to make up for some of the expenditures will be lost). As it can be inferred, if during a certain period of time there is more energy sold than the one consumed, electricity cost will result negative for the V2G, which means that the owner of the vehicle will be obtaining a profit from trading with the electricity, rather than just reducing its costs via V2G usage. Both equations have included the inflation rates for ICE fuel and electricity (*Inff*). Lastly, since according to [17] there will be 95% efficiency when charging a vehicle via plug-in charging mode, an efficiency factor ($e_f$) has been introduced to reflect the small loss of charge when energy is transferred in and out of the electric vehicle.

$$F_{ICE} = \sum_{i=1}^{j} (1 + Inff)^i \times (Av_{consICE} \times Cf_{ICE}) \tag{12}$$

$$F_{V2G} = \sum_{i=1}^{j} (1 + Inff)^i \times \left( E_{buy} \times C_{rpbuy} \times e_f - E_{sell} \times C_{rpsell} \times e_f \right) \tag{13}$$

The cornerstone of the vehicle-to-grid technology is the capability to sell electricity to the power grid where it is installed, since it offers a unique selling point that cannot be found in other regular vehicles. Thus, the energy that can be sold back to the system has been accounted in (14). If a yearly period is considered regarding the energy that can be sold ($E_{sell}$), then the overall available energy to trade—that is to say, energy that can be sold during peak hours, as opposed to the most advisable time to purchase it, which would be valley hours—will be the remaining energy after considering two variables from all the energy that has been bought for charging the battery ($E_{buy}$): a) the energy consumed to move the automobile ($E_{cons}$) and b) the passive discharge of the battery when it is idle (*pdis*). Yearly amount of energy sold and bought from the power grid can be considered after learning past patterns in energy pricing and consumption. Information for a long-time span can be found from the transport system operator if required [38].

$$E_{sell} = E_{buy} - E_{cons} - pdis \tag{14}$$

In order to understand the previous equation, $E_{buy}$ and $E_{cons}$ must be defined too. The energy that is bought for the battery of the V2G will result from calculating the amount of power ($Pw$) purchased during a certain period of time ($t$). However, the degradation of the battery will take its toll during the battery lifetime, resulting in declining energy storage capabilities. In addition, the passive discharge of the battery must also be born in mind. While it is negligible in the short term, its effects are more noticeable during the whole lifespan of the battery. Lastly, the difference between the nominal and the actual battery charge values must also be considered. These two latter variables are hard to quantify and no work from the literature seems to portray them in an accurate manner in mathematical models for V2G technology. As far as the V2G model is concerned, they have been included as *Dg* (degradation factor for the battery). When numerical values are used to evaluate the model, the maximum discharge speed of the battery will also have to be considered as a non-functional requirement, as no battery can provide an immediate amount of limitless energy. Due to this, *Dg* will have a role in the model, even though differences may not be that significant according to D. Wang et al. [39] or H. Ribberink et al. [40]. Battery degradation for purchased energy has been included in (15).

$$E_{buy} = \sum_{i=1}^{j} (Pw \times t)_i \times (1 - Dg) \tag{15}$$

Battery degradation has been estimated by the authors of this manuscript to be at 1.25% of its total capacity per year so it can be included with more accuracy in the mathematical model. The reviewed literature shows extreme disparity regarding this value, with some sources claiming that it will degrade up to 10% after 160,000 miles for an electric vehicle [41]. However, battery degradation considered for this scenario has been regarded as significantly higher, as a) suboptimal charge and discharge behavior patterns from the vehicle owners must be taken into account, and b) V2G usage of an electric vehicle implies a heavier utilization of the vehicle battery. A more realistic approach is found in [42], where a thorough V2G-based experiment was run with experimental lithium batteries showing that they would reach their end of life (EOL), regarded to be the point when the battery has lost 20% of its original maximum capacity retention, after 3000 cycles of charge and discharge. For the purpose of this mathematical model, it has been estimated that, on average, 1000 cycles will take place every year for the V2G solution (as described in [43]), and after eight years the battery total capacity will be depleted a 20% and have to be replaced with a new one. Thus, battery degradation is defined as represented in (16).

$$Dg = 0.0667 \times \sum_{i=1}^{j} i \tag{16}$$

The energy that is consumed by the V2G solution can be described as the average energy consumption of the vehicle during a specific distance ($E_{cons}$). As explained before, passive energy losses have been included as the *pdis* parameter.

### 3.3. Cost of Maintenance

Although it is not bound to happen inevitably, the battery used in the EV-V2G is very likely to eventually have to be replaced. However, it does not necessarily mean that the vehicle owner will pay for the full replacement if the vehicle has been acquired under a battery leasing agreement. Therefore, there are two possible options: if the vehicle and the battery are purchased, battery replacement costs will have to be considered; with the technology available today in commercial products, it is unlikely that a vehicle battery will outlive the vehicle itself. The other option, though, is that the vehicle manufacturer leases the batteries to the vehicle owner during a certain time period. In this way, battery reposition could be regarded as a periodic payment ($Bleas_i$) done during the lifetime of the vehicle. This latter scenario is modelled in (17) as $Bleas_i$; while this is not the default choice for consumers buying an electric vehicle, it is a feature usually overlooked in other models for V2G, so it has been included in this analysis. When price data are used to estimate the cost differences between acquiring and leasing the battery in the V2G, $Bleas_{tot}$ would be used as the maintenance cost for rented batteries, whereas $Batr$, added in (19), will be used as the parameter representing the cost of a battery replacement when the battery is purchased with the EV.

$$Bleas_{tot} = \sum_{i=1}^{j} Bleas_i \tag{17}$$

Lastly, maintenance costs have been included in the model as a way to evaluate the differences between the two kinds of vehicles. The ICE vehicle (18) makes use of a maintenance rate ($Drate_{ICE}$), in a way that resembles the one presented in [9], but using private transport rather than a school bus. Labor costs of refilling the fuel ($Lab$) and distance ($D$) have also been included. Furthermore, the equation conceived for the V2G solution (19) is making use of an equivalent rate ($Drate_{V2G}$) and a distance $D$ and the cost of one battery replacement ($Batr$). Taking into account the average lifetime of EV vehicles and of their batteries before a replacement (which can be estimated at roughly eight years according to the period warranty used in most car manufacturers [43,44]), it is more likely that a new vehicle will be acquired rather than a new whole battery is bought more than once. The equation that has been added as (19) can be modified to consider how battery costs impact the maintenance of a V2G solution when the battery is leased instead of purchased (20). Note that both kinds of vehicles will require the payment of yearly insurance costs (which has been represented by $Ins$). However, according to [45], their payment can be regarded to be the same for them.

$$M_{ICE} = \sum_{i=1}^{j} \left( Drate_{ICE} \times D + Lab + Ins \right)_i \tag{18}$$

$$M_{V2G} = \sum_{i=1}^{j} \left( Drate_{V2G} \times D + Ins \right)_i + Batr \tag{19}$$

$$M_{V2G} = \sum_{i=1}^{j} \left( Drate_{V2G} \times D + Ins + Bleas \right)_i \tag{20}$$

## 4. Numerical Assessment

The equations of the mathematical model described previously have been put to use for three different use cases, namely, professional drivers (that is to say, people that drive as a way to make their living), frequent drivers (people that drive on a usual basis), and occasional drivers (people that

drive rarely), under certain considerations and assumptions as described in the following subsections. Most references and subsidy figures that have been used are relative to the United States of America, due to the fact that it is one of the places where the amount of information was plentiful enough to obtain the data used in this study. Specifically, data for professional drivers was very reliable as it was based on statistics from taxi drivers that are offered online freely. The definition of these use cases is pivotal for the study that has been carried out, as the usage that is done of the V2G solution differs greatly in each of them. Depending on the usability of the vehicle for travelling, V2G capabilities will become prominent. For example, the greater amount of distance that a V2G solution works, the lower energy will be left to trade it when it is suitable.

### 4.1. Considerations

Table 3 contains the information regarding how the variables that have been introduced in the previously detailed equations have been given numeric values according to the existing related work. Some of those variables do not change in the three scenarios but many other do so, as they are closely linked to the case study involving the vehicle (fuel, distance driven, etc.). In this manuscript, the price of a V2G solution has been estimated to be $7500 higher than an ICE-powered counterpart; as far as the United States are concerned, financial aid of up to that quantity is offered to the buyers of a full EV solution in some regions [3,46], so it has been included as an EV overprice in the model.

As for the battery replacement, it has been regarded as an average value of the figures found in [47] and [48]. The result has been depicted in Table 2, which considers four vehicle models. Several car models are considered in this chart, according to the information provided in [47]. It has been considered that the data in [47] can be divided into a best case scenario with a 40 kWh battery, where the Nissan Leaf owner does not require to pay any extra other than the battery replacement, and a worst case scenario where the Nissan Leaf owner must pay both for the special adapter kit ($225) and labor costs of $1000 when the old battery is exchanged with the new one (also with a 40 kWh battery). These results demonstrate alignment with other studies that show battery cost to have been declining during the last decades, such as the one shown in [49].

**Table 2.** Average battery cost per kilowatt/hour.

| Vehicle | Battery Cost ($) per Kilowatt/Hour | Reference |
|---|---|---|
| Nissan Leaf best case scenario | $5499/40 = 137.45 $/kWh | [47] |
| Nissan Leaf worst case scenario | ($5499 + $1000 + 225)/40=168.1 $/kWh | [47] |
| Chevrolet Bolt EV | 205 $/kWh | [48] |
| Tesla Model 3 | 190 $/kWh | [48] |
| **Average** | **175.14 $/kWh** | N/A |

Considering that a vehicle battery of 40 kWh has been used for this manuscript, the cost of its replacement used in the numerical assessment results in **175.14 $/kWh × 40 kWh = $7005.6**.

It must be noted that the figures corresponding to professional, frequent, and occasional drivers are strongly related to the information that has been inferred from several sources present in this manuscript, such as [50] and [58]. It is said in [50] that taxi cabs can be driven up to 70,000 miles, whereas it is claimed in [58] that average miles travelled by a vehicle are 11,370. This is the mileage that has been defined for frequent drivers (people who drive a car often enough to require it during a significant amount of days of the year but do not make a living out of using automobiles). In order to strengthen the criteria used to have an accurate view of the mileage that defines each case study (professional, frequent, and occasional drivers) two more references have been studied. On the one hand, it is said in [66] that 2813 gallons per car and per year are consumed by taxi drivers, who represent the archetypical professional driver use case. On the other hand, it is claimed in [67]

that 524 gas gallons are used yearly per vehicle. Despite these figures are prone to change as time goes by or depending on boom or bust economic cycles, they can be used as representative values of mileage and gas consumption. Consequently, and considering the ratio of gas usage existing between professional and frequent drivers (2813/524 = 5.368) it has estimated that a) since frequent drivers drive 11,370 miles per year and b) mileage figures for professional drivers are unlikely to go beyond 70,000 miles per year, professional driver mileage can be estimated as 11,370 × 5.368 = 61,033 miles per year. As it will be described in use case C, due to the data presented in [68], it has been estimated that occasional drivers make use of automobiles a quarter of time (which has been correlated to mileage) than frequent drivers. Another aspect to consider is the relationship between the mileage for each use case and the energy being used in every one of them. It has been estimated that, according to the figures that can be obtained from [50] and [58] and the ratio of gas usage explained in the previous paragraph, yearly mileage will be of 61,033 miles for a professional driver, 11,370 miles for a frequent driver and 2842.5 miles for an occasional driver. Additionally, if the average figures that can be extracted from [61] are considered as well, battery consumption would be of 20,301.67 kWh for professional drivers, 3781 kWh for frequent ones and 945 kWh for occasional ones. Furthermore, there is a certain battery degradation coming from using the V2G functionalities of the enhanced EV which is far more significant than usual wear off in an EV battery. Consequently, the energy that can be traded every year depends on (a) the amount of energy available for trade (the more frequent a person drives, the higher amount of energy is used for driving; hence, V2G energy costs will be overall higher as lower profits can be made from trading) and (b) battery degradation (as time goes by, capacity of the battery will shrink). These considerations are especially important for Tables A5–A7, where profitability of the solution is described in relation to whether battery degradation is present or not.

### 4.2. Case Study A: Professional Drivers

This case study involves people whose main job implies driving or taking passengers in a private-like means of transport (taxi drivers are the most typical example). This kind of job implies that there will be high costs in consumed fuel and maintenance for ICE-based vehicles. As represented in Table 3 and mentioned earlier, the costs and usage for professional drivers have been calculated considering those according to [53], namely, the yearly average consumption of gas is 2813 [63]/524 [67] = 5.368 times the one made by the frequent drivers even if, as mentioned before, there are cases where taxi cabs are driven up to 70,000 miles per year [45]. The following figures, adjusted to inflation, have been obtained:

$$Ex_{ICE} = \$68,843.39 \ Ma_{ICE} = \$404,203.13$$

$$F_{ICE} = \$90,926.12$$

It can be inferred that the total costs for a professional driver using an ICE automobile for twelve years are the following ones:

$$C_{totICE} = \$35,285 + \$563,972.65 = \$599,257.65$$

If a V2G solution is used instead of an ICE-based vehicle, the results obtained when adjusted to inflation are different and overall lower:

$$Ex_{V2G} = \$12,278.20$$

$$Ma_{V2G} = \$90,887.29 \text{ total (with a 40 kWh battery)}$$

$$Fc_{V2G} = \$ 19,602.71$$

$$V2G \ conversion + Cost \ of \ the \ installation = \$1936$$

From these figures, it is calculated that the total costs for a professional driver using a V2G automobile are:

$$C_{totV2G} = \$42,785 + \$1936 + \$7005.60 + \$12,278.20 + \$19,602.71 + \$90,887.29 = \$174,494.79$$

**Table 3.** Variables included in the mathematical model.

| Variable | Description | Value (ICE) | Value (V2G) |
|---|---|---|---|
| $Av_{consICE}$ | Yearly average consumption of gas to move the ICE vehicle | 2813 */524 **/131 *** gallons [50] | – |
| $Avc_{buy}$ | Average cost of bought energy | – | 9.35 cents/kWh (off-peak hours) [51] |
| $Avc_{sell}$ | Average cost of sold energy | – | 15 cents/kWh (peak hours) [51] |
| $Batr$ | Battery replacement | – | $7005.60 [47,48], Table 2 |
| $Bleas$ | Yearly battery leasing | – | ca. $140 × 12 [52] |
| $Bleas_{tot}$ | Total cost of battery leasing | – | $Bleas$ × 12 |
| $Capex_{ICE}$ | Cost of acquiring an Internal Combustion Engine-powered vehicle | $35,285 [53] | – |
| $Capex_{V2G}$ | Cost of acquiring an Vehicle-to-Grid-powered vehicle | – | (2) |
| $C_{ICE}$ | Yearly carbon dioxide emissions for an Internal Combustion Engine vehicle | 19.6 (8.89 kg) lbs/gallon [54] | – |
| $C_{V2G}$ | Yearly carbon dioxide emissions for a Vehicle-to-Grid-powered vehicle | – | 149.25 [55] × 40 = 5,97 MT/12 years = 497 kg/year |
| $Cf_{ICE}$ | Cost of the fuel for an Internal Combustion Engine-powered vehicle | $2.176/gallon [56] | – |
| $C_{EV}$ | Cost of acquiring the Electric Vehicle | – | $42,785 [53,57] |
| $C_{rpbuy}$ | Real price of bought energy | – | (12) |
| $C_{rpsell}$ | Real price of sold energy | – | (13) |
| $C_{totICE}$ | Total costs of purchase and usage of the Internal Combustion Engine vehicle | (1) | |
| $C_{totV2G}$ | Total costs of purchase and usage of the Vehicle-to-Grid automobile | | (2) |
| $D$ | Yearly distance | 61,033 */11,370 **/ 2842.5 *** miles [58] | 61,033 */11,370 **/ 2842.5 *** miles [58] |
| $Dg$ | Degradation factor of the battery (State of Health) | – | (16) |
| $Drate_{ICE}$ | Average maintenance rate per mile by an Internal Combustion Engine-powered vehicle (medium sedan) | $0.5762−$0.116 [59] = $0.4602 | – |
| $Drate_{V2G}$ | Average maintenance rate per mile by a Vehicle-to-Grid-powered vehicle | – | $0.09204 (1/5 of [9,59]) |
| $E_{buy}$ | Yearly amount of energy bought | – | 2304 kWh × 1000 battery cycles (from [60]) |
| $E_{cons}$ | Yearly Amount of energy consumed | – | 20,301.67 */3781 **/ 945 ***kWh [61] |

**Table 3.** *Cont.*

| Variable | Description | Value (ICE) | Value (V2G) |
|---|---|---|---|
| $e_f$ | Efficiency factor for charging a vehicle via power cable | – | 0.95 [17] |
| $E_{sell}$ | Yearly amount of energy sold | – | (13) |
| $EX_{ICE}$ | Externalities for an Internal Combustion Engine-powered vehicle | (8) | – |
| $EX_{V2G}$ | Externalities for a Vehicle-to-Grid (V2G)-powered vehicle | – | (9) |
| $F_{ICE}$ | Gas costs for an Internal Combustion Engine-powered vehicle | (12) | – |
| $F_{V2G}$ | Energy costs for a Vehicle-to-Grid vehicle | – | (13) |
| $f_{bb}$ | Factor for energy purchase in optimal buying hours | – | 0.8 ***** |
| $f_{bs}$ | Factor for energy purchase in optimal selling hours | – | 0.2 ***** |
| $f_{sb}$ | Factor for energy sell in buying optimal hours | – | 0.8 ***** |
| $f_{ss}$ | Factor for energy sell in selling optimal hours | – | 0.2 ***** |
| $h_{ICE}$ | Per-mile cost of the health impact caused by the electricity consumed by the ICE vehicle | $0.07 (estimated from [9,10]) | – |
| $h_{V2G}$ | Per-Mile cost of the health impact caused by the electricity consumed by the V2G vehicle | – | $0.0149 [9] |
| $Heq$ | Cost of the installation of the required equipment to charge the Electric Vehicle | – | $1200 [61] |
| $Inf$ | Average inflation 2008−2019 (US) | 1.76% [62] | 1.76% [62] |
| $Inff$ | Inflation rate on fuel | 3.8% [57] | 1.9% [9] |
| $Ins$ | Yearly cost of insurance | $1251 [58] | $1251 [58] |
| $Lab$ | Yearly fuel refill labor | $1207.80 */225 **/56.25 *** [9] | – |
| $M_{ICE}$ | Maintenance costs of an Internal Combustion Engine-powered vehicle | (18) | – |
| $M_{V2G}$ | Maintenance costs of a Vehicle-to-Grid vehicle | – | (19) |
| $Opex_{ICE}$ | Operational costs to have the ICE vehicle in fully working condition | (4) | – |
| $Opex_{V2G}$ | Operational costs to have the V2G vehicle in fully working condition | – | (5) |
| $pdis$ | Passive discharge of the battery | – | 5.59%/30% **/ 120% **** [63] |
| $Pw$ | Amount of power purchased | – | $Ebuy/t$ |
| $SCC$ | Social Cost of Carbon | $37.20/MTCO2e ([64], calculated for 2016 dollars) | $37.20/MTCO2e ([60], for 2016 dollars) |
| $t$ | Period of time | – | Variable; 12 years for Section 5 |
| $Vcons$ | Cost of conversion to V2G technology | – | $736 [65] |

\* Professional drivers, \*\* Frequent drivers, \*\*\* Occasional drivers \*\*\*\* 120% represents that a charge cycle and a fifth of another one are lost \*\*\*\*\* Chosen as a plausible hypothesis.

As it can be inferred from the previous calculations, it can be seen that the V2G solution is far more economically efficient for a professional driver in the long term than an ICE vehicle. The graphical representation of the cumulative result that has been calculated for each of the years is displayed in Figure 2. At the same time, Table A1 is showing in the Appendix A how numerical calculations vary on a yearly basis as well.

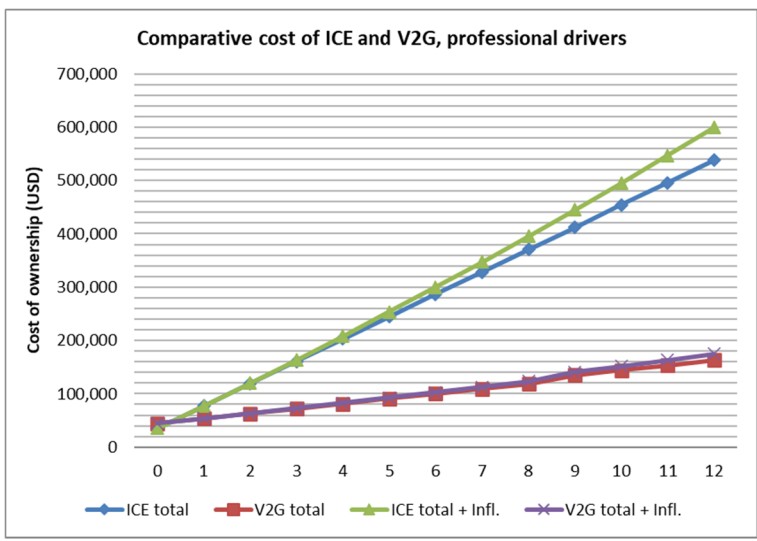

**Figure 2.** Graphical representation of the calculation results for professional drivers.

If the results that have been obtained are separated so that operational costs can be considered more accurately, it can be seen how despite a) a higher Capex if an EV is purchased; b) the required infrastructure to make the EV work as a V2G solution; and c) a battery renewal, the operational costs of the ICE vehicle are far higher in this scenario starting from year 1, mostly but not only, due to the maintenance costs required to have the ICE working satisfactory. This is the key advantage that the V2G solution has, which makes it economically far more advisable under these circumstances if compared to the ICE alternative. The graphical comparison of Opex costs has been displayed in Figure 3. Note that the battery replacement has been included as an Opex-related expenditure, so it is present in the cumulative figures. Moreover, it is considered that the first year of usage (year 0 in the previous graphs) there are not operational costs, which start being added at year 1. That is why this and the other equivalent graphs show year 0, whereas Opex-related ones do not.

Note that the previous results have been obtained under conditions deemed as "suboptimal" in terms of cost of energy purchase and sell. That is, all the energy has been bought during valley hours and sold during peak hours in a proportion of 80/20. This implies that according to the parameters that have been included in (8), (9), (10), and (11), it has been considered that 80% of the energy purchased was done so during valley hours and the other their during peak hours (so that $f_{bb} = 0.8$ and $f_{sb} = 0.2$), whereas 80% of the energy was sold during peak hours and the other 20% during valley ones (and thus, $f_{ss} = 0.8$ and $f_{bs} = 0.2$). Furthermore, small losses when charging and discharging the vehicle may result in a loss of electricity during these procedures (hence, $e_f = 0.95$, as described in Table 3. Lastly, the degradation of the battery has also been considered when doing the calculations according to the mathematical model. Hence, the energy that has been estimated to be sold every year decreases over time in the rate established in (14).

Overall, the proportion that is sold during each of the time periods will depend on the available power to operate in the market and the availability of the user of the V2G solution. There are two important aspects that can be inferred by all these calculations: under a time period of longer length than the one used here (12 years) the advantages of the V2G solution over the ICE one will be even more notorious as the ones portrayed in this time span, as cost differences between both of them are always unfavorable for the ICE vehicle. Additionally, a suboptimal scenario has little to no influence in the calculations done for both battery rental and acquisition, as the former one will become unfavorable in the long term, according to the figures obtained for battery rental that have been introduced in Section 4.5.

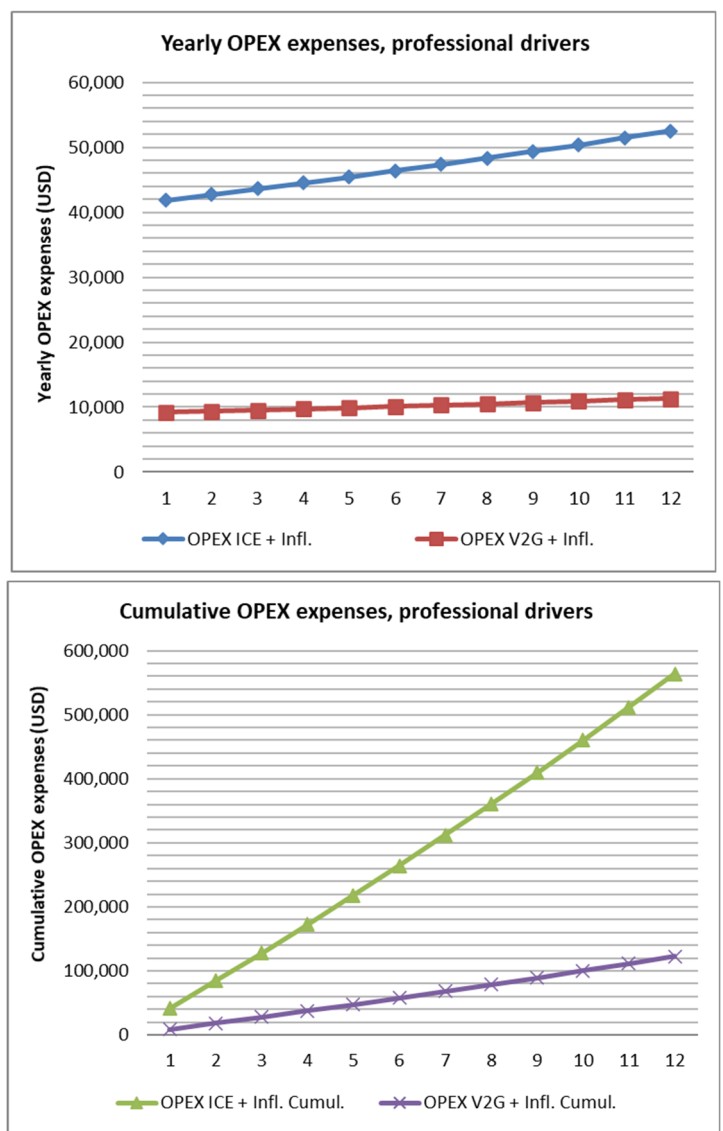

**Figure 3.** Yearly and cumulative OPEX expenses between an ICE vehicle and a V2G, professional drivers.

### 4.3. Case Study B: Frequent Drivers

The most representative situation that can be conceived for this use case is a freelance worker with a specific job that make them travel a significant distance every day (self-employed positions, etc.), but do not use driving as their business core. Frequent drivers are regarded in this numerical assessment as the average group of people, so they have been assigned the default figures that have been found in literature.

If the same calculations that were done previously are repeated for this use case, the next results are obtained adjusted to inflation:

$$Ex_{ICE} = \$12,824.83 \ Ma_{ICE} = \$88,770.04$$

$$F_{ICE} = \$16,937.54$$

Thus, the following costs will have to be assumed by the owners of an ICE vehicle during its lifetime will be

$$C_{totICE} = \$35,285 + \$118,532.40 = \$153,817.40$$

Should a V2G solution be used, the results would be

$$Ex_{V2G} = \$2486.41$$

$$Ma_{V2G} = \$30,401.65 \text{ (with a 40 kWh battery)}$$

$$Fc_{V2G} = \$ -9233.58$$

$$V2G \; conversion \; + \; Cost \; of \; the \; installation \; = \; \$1936$$

Note that the fuel (electricity) cost for the V2G vehicle is negative for this case study, due to the fact that selling the energy surplus is creating a profit for the end users of the vehicle, to the point that energy trading results economically advantageous for the end user in terms of energy costs. This is due to the fact that energy is being bought and sold in a proportion that makes the sold energy more economically significant in absolute values than the one that is being bought. Therefore, the engagement in energy trading for clients using the V2G solution becomes profitable, as the usage of vehicle-to-grid technology makes possible decreasing the costs of using an electric vehicle when the unused power is sold back. Thus, the total costs for a frequent driver that owns a V2G automobile would be:

$$C_{totV2G} = \$42,785 \; + \; \$1936 \; + \; \$7005.60 \; + \; \$30,401.65 \; - \; \$9233.58 \; + \; \$2486.41 \; = \; \$75,381.09$$

Figure 4 depicts the graphical representation of the obtained results whereas the calculations that have been carried out are presented in Table A2 of the Appendix A. Note that Figures containing graphs show a sudden non-linearity for the costs of the V2G solution event between years 8 and 9. This is due to the fact that it has been estimated that it will be the moment when battery from the V2G solution will have to be eventually replaced, so expenses rise accordingly to the $7005.60 that have to be spent. Moreover, benefits towards the V2G solution do not start right away but after year 2, thus showing that this scenario is less advantageous due to the lower costs for the ICE vehicle.

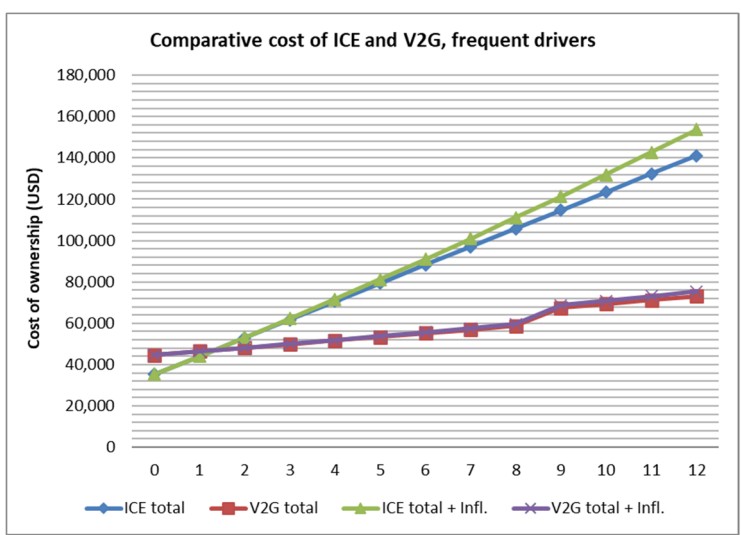

**Figure 4.** Graphical representation of the calculation results for frequent drivers.

As mentioned previously, Figure 5 depicts the differences in operational costs between the V2G and the ICE vehicle. Albeit with a smaller gap resulting from the lesser usage of the automobiles, the results are essentially replicated for this case study: yearly expenses, and cumulative ones when the battery replacement costs are included, are lower for the V2G than the ICE vehicle.

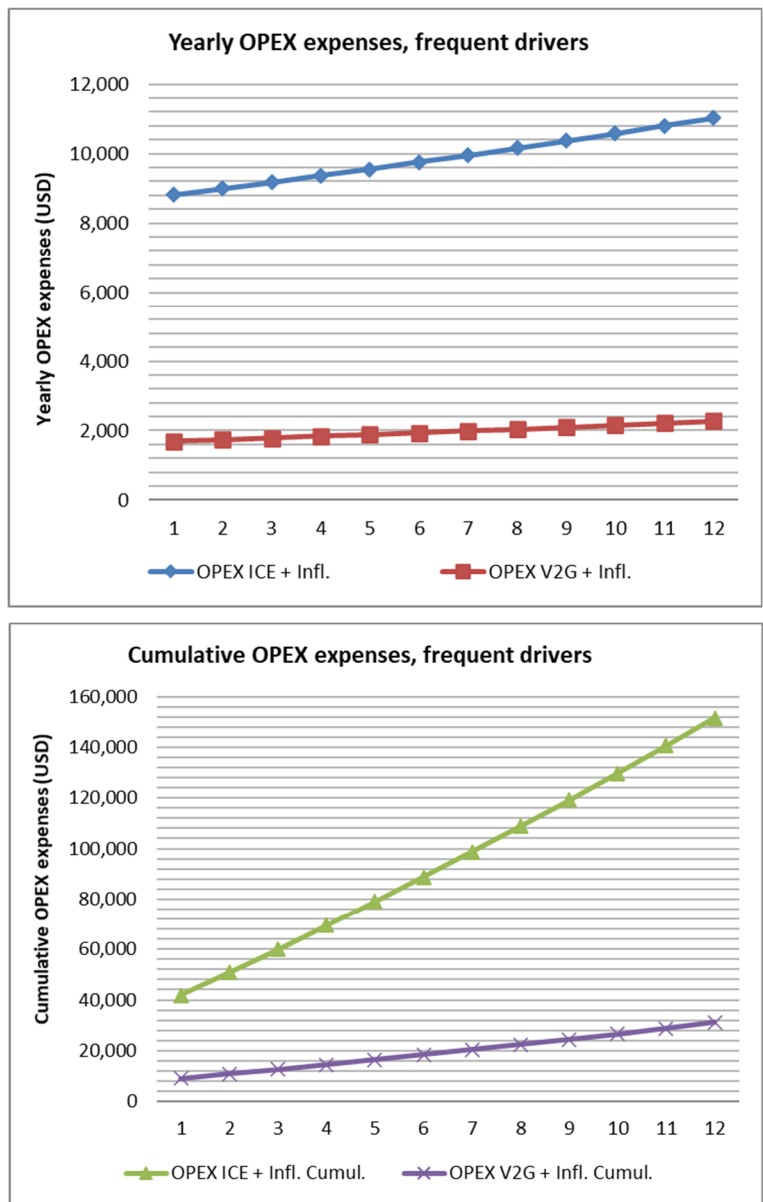

**Figure 5.** Comparison between yearly and cumulative OPEX expenses between an ICE vehicle and a V2G, frequent drivers.

As stated previously, purchasing and ICE vehicle results in a worse economy cost for the prosumer if compared to acquiring a full V2G solution (as the former implies higher costs of fuel, maintenance and externalities during the vehicle lifetime). Fuel consumption falls considerably for the V2G in this scenario, as more energy is used for trading operations. The suboptimal scenario is also used for this use case with the same set of variable values that was employed before ($f_{bb}$ =0.8, $f_{sb}$ =0.2, $f_{ss}$ =0.8, $f_{bs}$ =0.2, and $e_f = 0.95$). As in the previous case, despite obtaining a worse result with a suboptimal scenario where energy is neither bought nor sold under the best possible circumstances, it is still better than the one that would be obtained with the ICE solution.

*4.4. Case Study C: Occasional Drivers*

An occasional driver has been defined with the same criteria that was done in [68] and [69], that is to say, "A driver who operates a vehicle less than 25 percent of the total miles put on the car during a year". Consequently, it can be assumed that, when compared to frequent drivers, an occasional driver will use the vehicle one fourth of the time a frequent driver would, so all the expenses have

been considered to be one fourth of the ones calculated in the previous case study. As far as the ICE automobile is concerned, results adjusted to inflation are as follows:

$$Ex_{ICE} = \$3206.21 \; Ma_{ICE} = \$34,607.92$$
$$F_{ICE} = \$4234.38$$

Therefore, the resulting budget for an ICE vehicle owned by an infrequent driver would be

$$C_{totICE} = \$35,285 + \$42.048,52 = \$77,333.52$$

Thus, operational costs have become lower than the purchase of the vehicle itself. If a V2G solution is used, results obtained are

$$Ex_{V2G} = \$805.70$$
$$Ma_{V2G} = \$20,015.83 \; (\text{with a 40kWh battery})$$
$$F_{V2G} = -\$14,128.02$$
$$V2G \; conversion + Cost \; of \; the \; installation = \$1936$$

As it happened before, the fuel costs for electricity in this case are negative. What is more, since there is more electricity available to be sold (as it is used to a lesser extent by the vehicle), saving costs become even more prominent than in the previous case study. The final costs would be as follows:

$$C_{totV2G} = \$42,785 + \$1936 + \$7005.60 + \$805.7 + \$20,015.83 - \$14,128.02 = \$58,419.48$$

As it was done in the previous cases, the suboptimal scenario has been used with the same set of variables ($f_{bb} = 0.8$, $f_{sb} = 0.2$, $f_{ss} = 0.8$, $f_{bs} = 0.2$, and $e_f = 0.95$) energy costs are higher than in the optimal scenario. Unlike previous case studies, the EV-V2G is not as in clear advantage over the ICE vehicle in terms of expenses as it was before. What is more, it would not be until the fourth year of ownership that the V2G solution shows a better performance when compared to the ICE automobile. The main reason for this is that, although the V2G solution decreases its costs as long as the battery is kept the same, as soon as the latter is replaced, costs rise above the ICE level, thus closing the gap between the two kinds of vehicles. The graphical representation of this fact is shown in Figure 6.

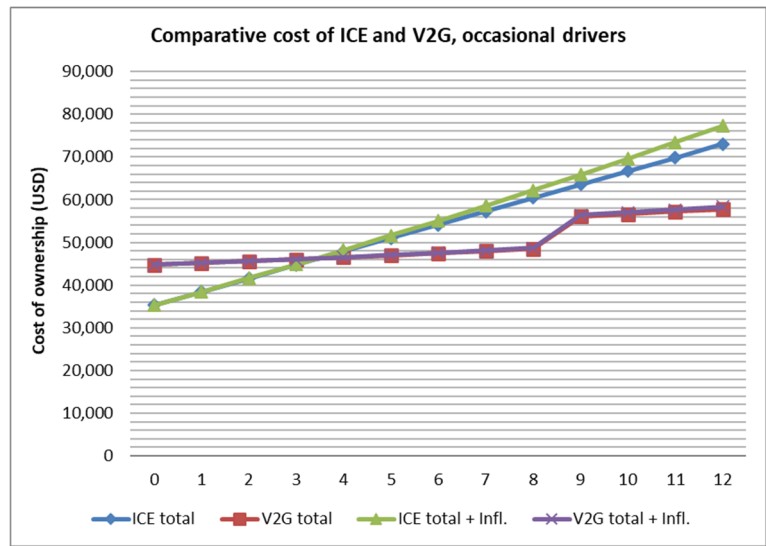

**Figure 6.** Graphical representation of the calculation results for occasional drivers.

Additionally, Figure 7 shows a comparison between operational costs between the V2G and the ICE options for occasional drivers. As in previous cases, yearly and cumulative expenses for operational costs are lower when the EVV2G is used instead of the ICE. However, the differences are less significant this time, to the point that the higher purchase cost of the EVV2G and its frequent battery replacement make it harder to justify using it. Interestingly enough, if the battery replacement is not taken into account, OPEX for the V2G shows almost stagnant figures. This is due to the fact that the V2G is used so little that it is highly available to trade energy in favorable terms with the overall grid system, and it results in a profit for the end user who owns it.

Table 4 shows a numerical summary of the cost of these three use cases.

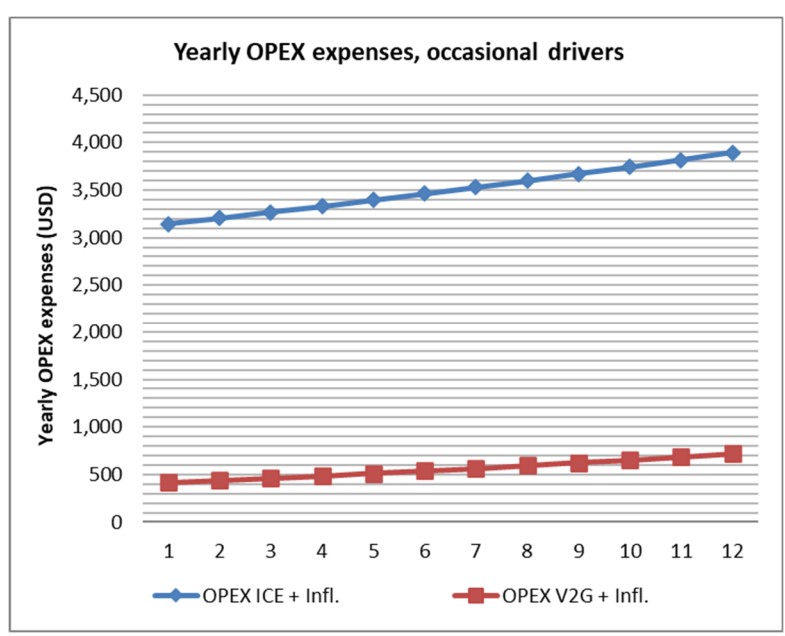

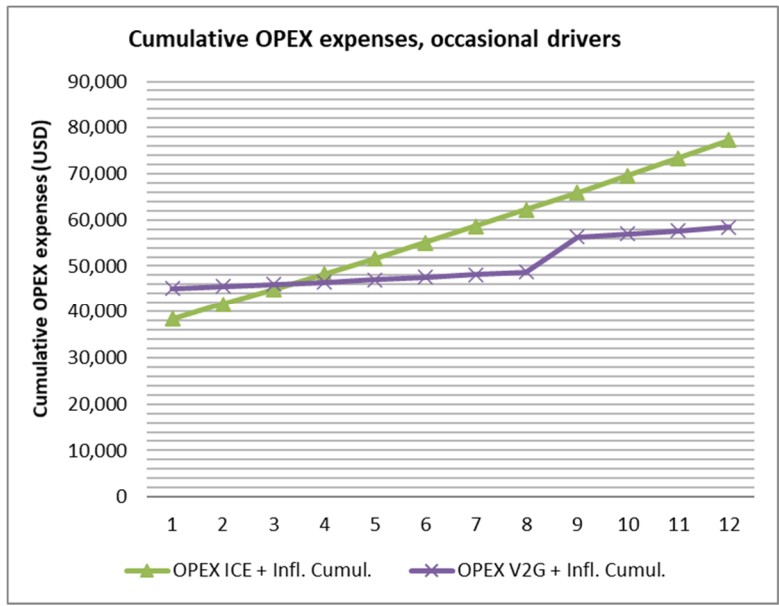

**Figure 7.** Yearly and cumulative OPEX expenses between an ICE vehicle and a V2G, frequent rivers.

**Table 4.** Costs summary of ICE and V2G.

|  |  | **Professional Drivers** | **Frequent Drivers** | **Occasional Drivers** |
|---|---|---|---|---|
| Externalities | ICE | $68,843.39 | $12,824.83 | $3206.21 |
|  | V2G | $12,278.20 | $2486.41 | $805.70 |
| Maintenance | ICE | $404,203.13 | $88,770.04 | $34,607.92 |
|  | V2G | $90,887.29 | $30,401.65 | $20,015.83 |
| Fuel | ICE | $90,926.412 | $16,937.54 | $4234.38 |
|  | V2G | $19,602.71 | −$9233.58 | −$14,128.02 |
| CAPEX | ICE | $35,285 | $35,285 | $35,285 |
|  | V2G | $44,721 | $44,721 | $44,721 |
| Total | ICE | $599,257.65 | $153,817.40 | $77,333.52 |
|  | V2G | $174,494.79 | $75,381.09 | $58,419.48 |

*4.5. Comparison between Battery Rental and Battery Ownership*

If it is chosen to purchase an EV where the battery is rented rather than acquired with the same vehicle, the average costs obtained after twelve years (adjusted to inflation) with one battery replacement according to the mathematical model are as follows:

$$CAPEX_{V2G} = \$51,726.60 \text{ (with a 40 kWh battery purchase)}$$
$$CAPEX_{V2G} = \$56,586.66 \text{ (with 40 kWh battery rental)}$$

The yearly comparison of each option has been depicted in Figure 8, whereas the calculations themselves have been placed in the Appendix A, Table A4.

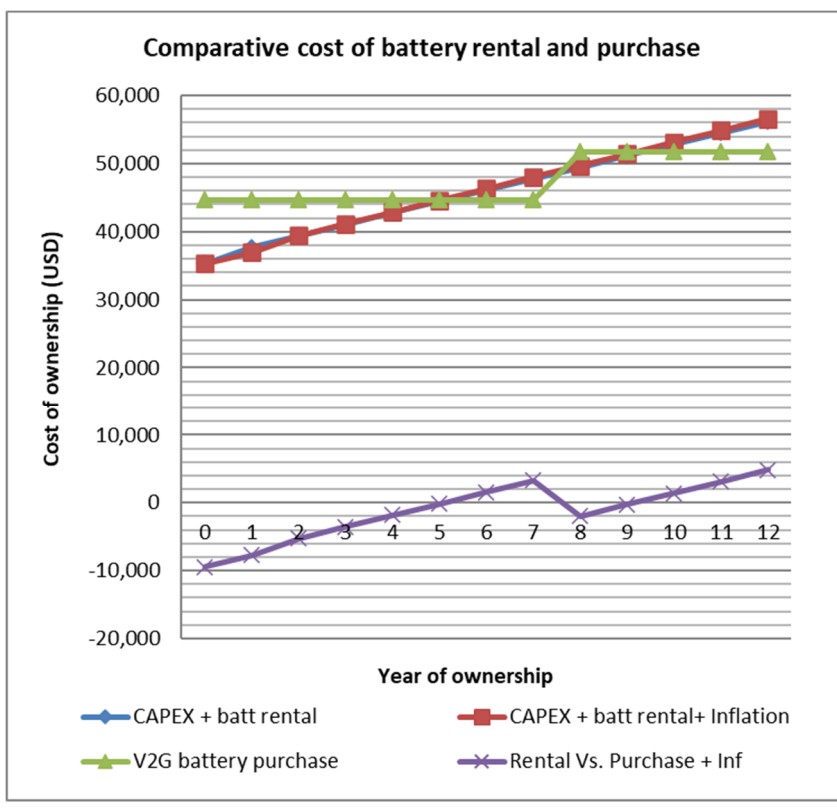

**Figure 8.** Graphical representation of battery rental costs versus battery purchase ones.

If these results are compared thoroughly, it can be seen that the battery purchase option becomes more advisable to use in the long term when the EV has been enabled to make use of V2G technology, whereas it is the opposite for shorter term ownership (5 or less years). This is due to the fact that the battery installed in the vehicle becomes depleted at a faster rate than a conventional EV which makes no use of V2G, thus being more likely to have its battery replaced once during its usage timespan. Should the battery not require to be replaced, then battery ownership option would result more competitive. However, it must be taken into account that usually, no manufacturer that offers battery rental as an option expects its customers to use it as part of the equipment of V2G technology. Probably, manufacturers would put restrictions to their usage if end users were openly planning to use their vehicles with this kind of technology.

It must be noted that, with the battery rental option, all the considerations done previously regarding battery degradation are still valid. Battery will degrade at a similar rate regardless of how the owner pays for its usage, as the components and chemical reactions that make it work remain the same in both cases. That is why battery replacements are considered under the purchase option, as any V2G solution will make use the car battery intensively (due to the very nature of V2G, which demands more frequent energy discharges and recharges than a EV battery used one-way only) and will have to be replaced after a relatively short amount of time, whereas a rented battery will degrade with the same parameters but the cost of its replacement will not have to be assumed by the end user.

Nevertheless, the scenario where the rented battery of the EVV2G is replaced every year could be put forward as another part of this study. In this case, yearly degradation could be considered as zero (as the battery would be replaced every year) and greater amounts of energy could be purchased and sold, due to the battery capacity being maintained during the lifetime of the V2G solution. In this case, more energy would be available for trading operations, thus resulting in an increase of the profitability of the rented battery V2G solution. Yearly surplus in energy availability would progressively increase compared to a purchased battery, as shown in Table 5. Should it be assumed that only yearly degradation is taking place with the battery rental option, results would vary in favor of the latter, but the overall tendency would be the same: for long periods of time, battery purchase would be more efficient than battery rental.

**Table 5.** Profit difference between battery degradation and non-battery degradation. Assuming a yearly replacement of the battery, it would be the difference of battery renting vs. battery ownership.

| Year | Difference Degree/No Degree | Difference Purch./Rental No Deg. |
|---|---|---|
| 1 | 0.00 | −9436 |
| 2 | 16.40 | −7726.43 |
| 3 | 33.42 | −5282.34 |
| 4 | 51.05 | −3588.75 |
| 5 | 69.29 | −1895.22 |
| 6 | 88.14 | −201.75 |
| 7 | 107.60 | 1491.66 |
| 8 | 127.68 | 3185.03 |
| 9 | 148.36 | −2127.24 |
| 10 | 169.66 | −433.92 |
| 11 | 191.57 | 1259.37 |
| 12 | 214.10 | 2952.67 |

The yearly figures that would be obtained would be as portrayed in Tables A5–A7 for professional, frequent and occasional drivers. Note that regardless of the kind of driver that makes use of the

solution the difference is the same in every case, as the increase in energy available is due to the same reason (the same improvement in energy used for trade operations).

## 5. Impact on Grid Utilities

The previously described model has been conceived for its usage in V2G solutions that become part of the entities able to provide power to sell and purchase at the electricity markets. For example, in [70] it is stated that despite the dominant trend in charging V2G is using off-peak hours, coincident user patterns can pose a threat for power system components both when charging vehicles and injecting power to the grid. The authors claim that it would be possible to overcome that problem by assessing the suitable V2G penetration level for optimal operation and precisely planning the V2G behavior on the distribution system. Furthermore, [71] describes how the addition of V2G parking lot facilities creates additional energy losses in the feeders of the electric utility owners derived from the behavior of reactive power injection and the load patterns of the users. A way to minimize this issue would be locating optimally a parking lot along the aforementioned feeder.

Additionally, V2G technology can be used to make power consumption more regular and avoid the peaks and valleys that create issues for the power grid: because of the tendency of end users to charge their vehicles in off-peak hours and not to demand that energy during the peak ones, V2G effectively becomes a way to enhance peak shaving and valley filling curves of energy demand. It is stated in [72] that combining V2G solutions with energy storage and photovoltaic electricity generation could result in a reduction of up to 37% during peak periods. Furthermore, in [73] it is claimed that by following a strategy based on comparing a forecasted load curve with another one based on forecasting available charge and discharge power peak shaving can be controllable and real, thus proving that V2G can be used as a way to create a more balanced demand of electricity. What is more, it is said in [74] that a high penetration of EVs is very likely to demand a stronger and more reliable power network; according to the authors of that manuscript, transformer replacement costs reach 72% of the total deployed transformers value with an EV penetration of 50%. However, this manuscript does not consider that the added power for that V2G can be brought to the power grid. Moreover, it is stated in [75] that distribution transformer may experience a measurable loss of life resulting of the increased strain that power demanded by plug-in hybrid electric vehicles (PHEVs) may produce. In this case, this study deals mostly with how PHEVs interact with the grid, describing the possibility of using V2G technology as part of the applications of PHEVs. The results shown in this manuscript demonstrate that V2G can be a viable solution for end users to obtain an economical benefit with their vehicles. However, they also show that the status of development in batteries makes profitability difficult, as the rapid degradation and their relative expensive cost depletes most of the benefits that could be obtained. Arguably, other options implying reducing the costs of purchasing an EV and converting it into an EVV2G could be satisfactory, but such a solution looks unlikely to happen in the short term. Overall, it is assumed that the existence of V2G solutions will strain the power grid in the short term, but there are advantageous solutions that can be integrated in the resulting smart grid. A typical solution that could come to this system advantage would be the integration of V2G technology with the other components of the power grid via middleware architectures [76] so they can be seamlessly included in such heterogeneous deployments.

## 6. Conclusions

According to the study done in this manuscript, purchasing an EV or V2G automobile and having the battery leased, instead of bought altogether with the vehicle, is economically inefficient for periods of time longer than five years, except for a comparatively brief period of time after the vehicle battery is replaced. On the other hand, even if charging a V2G solution during long periods of time might not be a solution for some drivers and electricity has to be purchased sometimes during peak hours or sold during off-peak slots of time, V2G technology is still more economically efficient overall when compared to ICE vehicles. This can be seen in the results that have been obtained in the study

done, where it is estimated that a V2G becomes almost immediately more economically efficient for professional drivers, whereas the same happens for frequent drivers after one year. As for occasional drivers, it is estimated to take from 3 to 4 years for V2Gs to be more economically efficient than ICE-powered vehicles. Despite differences in the periods of time depending on the profile, there are two tendencies: a) the higher the usage of a vehicle, the faster it turns into a more economical choice to use V2G technology and b) the longer the time a V2G vehicle is used, the more convenient it is to buy the battery rather than renting it. Regardless of the more intense battery degradation and mandatory battery replacement that must be done in a V2G vehicle, it will result in a more economic usage in the medium-to-long term. Even if the battery had to be replaced two times in the timespan used in this study, results would still be favorable for V2G vehicles over ICE-powered ones.

That said, although V2G technology is more cost-efficient in the long term than ICE solutions, batteries are still the main bottleneck for greater profits, as they impose limits to the savings that can be done from purely maintenance costs. The potential profits (or at least, expenses reduction) that can result from applying this technology are strongly linked to battery degradation and battery costs. The first one puts a severe strain on the profitability of V2G as part of the appeal of acquiring an EV. According to the study done in this manuscript, the fact that batteries will have to be replaced in the EVV2G once limits the practical applicability of this technology to exploit it in a profitable manner. Battery costs, on the other hand, are expected to reduce over time based on the trend that has been taking place during the last fifteen to twenty years, and hence this variable will work in favor of V2G solutions as time goes by. Nevertheless, the cost of acquiring a vehicle usually becomes less important than the operational costs that have to be faced under a prolonged period of time, so the initial disadvantage of V2G-based solutions becomes far less significant in the long term. Furthermore, the system could be extended to public buildings and facilities, like parking lots, as long as the costs associated to that infrastructure made it worthy. Finally, according to the parameters used, the more a vehicle must be used, the more economically efficient V2G technology is. If a consumer is considering turning into a prosumer by means of a V2G solution, the mathematical model presented here is holistic enough to be applied to any other numeric values that the consumer may want to choose. If this latter idea is fully taken into account, the transition from a model made up by privately owned vehicles that are purchased and used during a very limited amount of time every day, to one based on vehicle sharing where many different users that do not own the vehicle use it almost continuously, becomes an alternative to consider.

Finally, should new challenges come up for sharing small spaces, such as the one represented by the COVID−19, sanitary protocols can be used to minimize the risk of infection to the greatest possible extent. While significant drops in car selling have taken place during the COVID−19 crisis [77], they are still usable, valid tools for transportation. Procedures used in public transport could be extended to car sharing, due to the resembling nature of all these use cases, such as (a) periodic vehicle sanitation, (b) periods of time for ventilation every timeslot a car is used, or (c) regular precaution measures that have become widespread during the pandemic (usage of disposable or washable masks and gloves) are but a few of the actions that could take place. There are several research lines that can be suggested as future works in safe usage of shared private and public means of transport, like applying COVID−19 AI-based prediction models [78] for EVV2G sharing.

**Author Contributions:** J.R.-M. provided the introduction, a significant part of the related works review, the foundations of the mathematical model and its numerical assessment, and the impact that the implementation of V2G would have in the power grids. P.C. made contributions to the equations of the mathematical model and the calculations derived from it, along with a review of the state of the art and the addition of some references to it. V.B. reviewed the whole manuscript and added notes to the parts that had to be improved in terms of quantity and quality. M.M.-N. participated in the overall readability of the manuscript, reviewed the mathematical model and cooperated in the creation of the graphs of the manuscript and the data tables. All authors have read and agreed to the published version of the manuscript.

**Funding:** This research received no external funding.

**Conflicts of Interest:** The authors declare no conflict of interest.

## Nomenclature

| Term | Meaning |
|---|---|
| Average distance rate | Used for maintenance and externalities measurement |
| Battery degradation | Used to calculate when the battery will have to be replaced |
| Battery leasing | Cost of leasing the battery |
| Battery replacement | Needed for continued energy storage |
| Capex | Cost of acquiring the vehicle |
| Carbon emissions | Quantity of carbon released by the vehicle |
| Cost of the Fuel | Evaluates difference between Internal Combustion Engine gas and electricity |
| Distance | Number of kilometres run by the vehicle |
| Energy consumption | Evaluates the consumed resources by the vehicle |
| Energy loses | Evaluates the loss of energy in the vehicle operation |
| Externalities | Impact in other areas related to the vehicle environment |
| Fuel consumption | Required to start and run the vehicle |
| Labour cost of battery change | Cost of a battery change |
| Labour cost of gas refilling | Cost of refilling gas |
| Maintenance | Costs used to keep the vehicle usable |
| Maximum real capacity | Actual capacity of the battery rather than the nominal one |
| Opex | Operational costs to keep the vehicle functional |
| Passive energy losses | Resulting from leaving the battery idle |
| Purchased energy | Electricity bought for the vehicle |
| Purchased power | Power bought during a certain amount of time |
| Revenues | Benefits from trading operations |
| Round-trip efficiency | Efficiency of energy usage in a full charge cycle |
| Social cost of carbon | Used to assess the impact on the environment by the vehicle |
| Sold energy | Electricity sold through the vehicle |
| Time | Amount of time for energy purchases |
| V2G conversion | Cost of turning an EV into a V2G |

## Appendix A

Tables and graphs representing calculations have been included in this manuscript in order to offer a clearer view of the change in costs and expenses over the amount of time used. To begin with, Table A1 shows how the calculations are done regarding the first scenario included in this manuscript, and how they increase each year. It can be seen that almost from the very beginning using a V2G vehicle is advantageous compared to an ICE one. Likewise, Table A2 shows how the costs evolve for frequent drivers during the same timespan that has been established for the other two scenarios. In a similar manner, Table A3 shows how calculations result for the use case of occasional drivers. All these figures have been calculated with and without adjusting them to inflation, so the impact of the expenditures under constant 2019 USD value and in a way that could evolve in the future can be shown. It has to be taken into account that although inflation overall has been regarded to have a similar impact on the ICE vehicle and the V2G solution, it is higher for fuel than for electricity, according to the historical data that has been retrieved. For the total costs combined, cumulative figures have been included in the two rightmost columns in each of the tables. Table A4 shows the increase in the expenditures for the V2G where the battery is either purchased or rented during the same period of time, whereas Figure 3 shows the graphical representation. It has been depicted how from the sixth year and on costs favor purchasing the battery of the vehicle instead of renting it, with the exception of the period of time taking place immediately after renewing the battery of the V2G vehicle. Finally, Tables A4–A7 show yearly difference in trading profitability depending on whether battery degradation is present or not. An example of this use case would be the yearly replacement of the battery in a V2G solution that is rented rather than purchased as part of the car.

**Table A1.** Cumulative costs for ICE and V2G solutions, professional drivers (figures are in USD).

| CAPEX ICE | Year | Externalities ICE | Ext. ICE w. Inflation | Maintenance | Maint. w. Inflation | Fuel | Fuel w. Inflation | OPEX ICE | OPEX ICE w. Inflation | CAPEX + OPEX ICE (Cumulative) | CAPEX + OPEX ICE w. Inflation (Cumulative) |
|---|---|---|---|---|---|---|---|---|---|---|---|
| 35,285 | | | | | | | | | | | |
| | 1 | 5202.59 | 5202.59 | 30,546.19 | 30,546.19 | 6121.09 | 6121.09 | 41,869.86 | 41,869.86 | 77,154.86 | 77,154.86 |
| | 2 | 5202.59 | 5294.16 | 30,546.19 | 31,083.80 | 6121.09 | 6353.69 | 41,869.86 | 42,731.64 | 119,024.73 | 119,886.51 |
| | 3 | 5202.59 | 5387.33 | 30,546.19 | 31,630.87 | 6121.09 | 6595.13 | 41,869.86 | 43,613.34 | 160,894.59 | 163,499.85 |
| | 4 | 5202.59 | 5482.15 | 30,546.19 | 32,187.58 | 6121.09 | 6845.74 | 41,869.86 | 44,515.47 | 202,764.46 | 208,015.32 |
| | 5 | 5202.59 | 5578.64 | 30,546.19 | 32,754.08 | 6121.09 | 7105.88 | 41,869.86 | 45,438.60 | 244,634.32 | 253,453.92 |
| | 6 | 5202.59 | 5676.82 | 30,546.19 | 33,330.55 | 6121.09 | 7375.91 | 41,869.86 | 46,383.28 | 286,504.19 | 299,837.19 |
| | 7 | 5202.59 | 5776.73 | 30,546.19 | 33,917.17 | 6121.09 | 7656.19 | 41,869.86 | 47,350.09 | 328,374.05 | 347,187.28 |
| | 8 | 5202.59 | 5878.40 | 30,546.19 | 34,514.11 | 6121.09 | 7947.13 | 41,869.86 | 48,339.64 | 370,243.92 | 395,526.92 |
| | 9 | 5202.59 | 5981.86 | 30,546.19 | 35,121.56 | 6121.09 | 8249.12 | 41,869.86 | 49,352.54 | 412,113.78 | 444,879.46 |
| | 10 | 5202.59 | 6087.14 | 30,546.19 | 35,739.70 | 6121.09 | 8562.58 | 41,869.86 | 50,389.42 | 453,983.65 | 495,268.88 |
| | 11 | 5202.59 | 6194.28 | 30,546.19 | 36,368.72 | 6121.09 | 8887.96 | 41,869.86 | 51,450.96 | 495,853.51 | 546,719.84 |
| | 12 | 5202.59 | 6303.30 | 30,546.19 | 37,008.81 | 6121.09 | 9225.70 | 41,869.86 | 52,537.81 | 537,723.38 | 599,257.65 |
| | Total | 62,431.08 | 68,843.39 | 366,554.24 | 404,203.13 | 73.453,06 | 90,926.12 | 502,438.38 | 563,972.65 | 537,723.38 | 599,257.65 |

| CAPEX V2G | Year | Externalities V2G | Ext. V2G w. Inflation | Maintenance | Maint. w. Inflation | Electr. | Elec. w. Inflation | OPEX V2G | OPEX V2G w. Inflation | CAPEX + OPEX V2G + Battery (Cumulative) | CAPEX + OPEX V2G w. Inflation + Battery (Cumulative) |
|---|---|---|---|---|---|---|---|---|---|---|---|
| 44,721 | | | | | | | | | | | |
| | 1 | 927.88 | 927.88 | 6868.48 | 6868.48 | 1387.16 | 1387,16 | 9183.52 | 9183.52 | 53,904.52 | 53,904.52 |
| | 2 | 927.88 | 944.21 | 6868.48 | 6989.36 | 1403.26 | 1429,92 | 9199.62 | 9363.49 | 63,104.13 | 63,268.01 |
| | 3 | 927.88 | 960.83 | 6868.48 | 7112.38 | 1419.36 | 1473,29 | 9215.72 | 9546,50 | 72,319.85 | 72,814.51 |
| | 4 | 927.88 | 977.74 | 6868.48 | 7237.55 | 1435.46 | 1517,28 | 9231.81 | 9732.57 | 81,551.66 | 82,547.08 |
| | 5 | 927.88 | 994.95 | 6868.48 | 7364.93 | 1451.55 | 1561,87 | 9247.91 | 9921.75 | 90,799.58 | 92,468.83 |
| | 6 | 927.88 | 1012.46 | 6868.48 | 7494.56 | 1467.65 | 1607,08 | 9264.01 | 10,114.10 | 100,063.59 | 102,582.93 |
| | 7 | 927.88 | 1030.28 | 6868.48 | 7626.46 | 1483.75 | 1652,90 | 9280.11 | 10,309.64 | 109,343.70 | 112,892.57 |
| | 8 | 927.88 | 1048.41 | 6868.48 | 7760.69 | 1499.85 | 1699,33 | 9296.21 | 10,508.43 | 118,639.90 | 123,401.00 |
| | 9 | 927.88 | 1066.86 | 6868.48 | 7897.27 | 1515.95 | 1746,37 | 9312.31 | 10,710.51 | 134,957.81 | 141,117.11 |
| | 10 | 927.88 | 1085.64 | 6868.48 | 8036.27 | 1532.05 | 1794,03 | 9328.41 | 10,915.93 | 144,286.22 | 152,033.05 |
| | 11 | 927.88 | 1104.75 | 6868.48 | 8177.71 | 1548.15 | 1842,29 | 9344.50 | 11,124.75 | 153,630.72 | 163,157.79 |
| | 12 | 927.88 | 1124.19 | 6868.48 | 8321.63 | 1564.25 | 1891,17 | 9360.60 | 11,337.00 | 162,991.32 | 174,494.79 |
| | Total | 11,134.56 | 12,278.20 | 82,421.73 | 90,887.29 | 17,708.43 | 19,602.71 | 111,264.72 | 122,768.19 | 162,991.32 | 174,494.79 |

**Table A2.** Cumulative costs for ICE and V2G solutions, frequent drivers (figures are in USD).

| CAPEX ICE | Year | Externalities ICE | Ext. ICE w. Inflation | Maintenance | Maint. w. Inflation | Fuel | Fuel w. Inflation | OPEX ICE | OPEX ICE w. Inflation | CAPEX + OPEX ICE | CAPEX + OPEX ICE w. Inflation |
|---|---|---|---|---|---|---|---|---|---|---|---|
| 35,285 | 1 | 969.19 | 969.19 | 6708.47 | 6708.47 | 1140.22 | 1140.22 | 8817.89 | 8817.89 | 44,102.89 | 44,102.89 |
| | 2 | 969.19 | 986.25 | 6708.47 | 6826.54 | 1140.22 | 1183.55 | 8817.89 | 8996.34 | 52,920.78 | 53,099.23 |
| | 3 | 969.19 | 1003.61 | 6708.47 | 6946.69 | 1140.22 | 1228.53 | 8817.89 | 9178.82 | 61,738.66 | 62,278.05 |
| | 4 | 969.19 | 1021.27 | 6708.47 | 7068.95 | 1140.22 | 1275.21 | 8817.89 | 9365.43 | 70,556.55 | 71,643.49 |
| | 5 | 969.19 | 1039.24 | 6708.47 | 7193.37 | 1140.22 | 1323.67 | 8817.89 | 9556.28 | 79,374.44 | 81,199.77 |
| | 6 | 969.19 | 1057.53 | 6708.47 | 7319.97 | 1140.22 | 1373.97 | 8817.89 | 9751.47 | 88,192.33 | 90,951.24 |
| | 7 | 969.19 | 1076.15 | 6708.47 | 7448.80 | 1140.22 | 1426.18 | 8817.89 | 9951.13 | 97,010.22 | 100,902.37 |
| | 8 | 969.19 | 1095.09 | 6708.47 | 7579.90 | 1140.22 | 1480.37 | 8817.89 | 10,155.36 | 105,828.10 | 111,057.73 |
| | 9 | 969.19 | 1114.36 | 6708.47 | 7713.31 | 1140.22 | 1536.63 | 8817.89 | 10,364.29 | 114,645.99 | 121,422.02 |
| | 10 | 969.19 | 1133.97 | 6708.47 | 7849.06 | 1140.22 | 1595.02 | 8817.89 | 10,578.05 | 123,463.88 | 132,000.07 |
| | 11 | 969.19 | 1153.93 | 6708.47 | 7987.20 | 1140.22 | 1655.63 | 8817.89 | 10,796.77 | 132,281.77 | 142,796,84 |
| | 12 | 969.19 | 1174.24 | 6708.47 | 8127.78 | 1140.22 | 1718.55 | 8817.89 | 11,020.56 | 141,099.66 | 153,817.40 |
| | **Total** | **11,630.28** | **12,824.83** | **80,501.69** | **88,770.04** | **13,682.69** | **16,937.54** | **105,814.66** | **118,532.40** | **141,099.66** | **153,817.40** |

| CAPEX V2G | Year | Externalities V2G | Ext. V2G w. Inflation | Maintenance | Maint. w. Inflation | Electr. | Elec. w. Inflation | OPEX V2G | OPEX V2G w. Inflation | CAPEX + OPEX V2G + Battery | CAPEX + OPEX V2G w. Inflation + Battery |
|---|---|---|---|---|---|---|---|---|---|---|---|
| 44,721 | 1 | 244.70 | 244.70 | 1559.96 | 1559.96 | −788.42 | −788.42 | 1016.24 | 1016.24 | 45,737.24 | 45,737.24 |
| | 2 | 244.70 | 249.40 | 1559.96 | 1589.91 | −772.33 | −787.01 | 1032.33 | 1052.30 | 46,769.57 | 46,789.55 |
| | 3 | 244.70 | 254.19 | 1559.96 | 1620.44 | −756.25 | −784.99 | 1048.41 | 1089.64 | 47,817.98 | 47,879.18 |
| | 4 | 244.70 | 259.07 | 1559.96 | 1651.55 | −740.17 | −782.36 | 1064.49 | 1128.26 | 55,888.07 | 56,013.04 |
| | 5 | 244.70 | 264.04 | 1559.96 | 1683.26 | −724.08 | −779.11 | 1080.58 | 1168.19 | 56,968.65 | 57,181.23 |
| | 6 | 244.70 | 269.11 | 1559.96 | 1715.58 | −708.00 | −775.26 | 1096.66 | 1209.43 | 58,065.31 | 58,390.66 |
| | 7 | 244.70 | 274.28 | 1559.96 | 1748.52 | −691.92 | −770.80 | 1112.74 | 1252.00 | 66,183.65 | 66,648.25 |
| | 8 | 244.70 | 279.54 | 1559.96 | 1782.09 | −675.84 | −765.72 | 1128.82 | 1295.91 | 67,312.47 | 67,944.17 |
| | 9 | 244.70 | 284.91 | 1559.96 | 1816.31 | −659.75 | −760.04 | 1144.91 | 1341.18 | 68,457.38 | 69,285.35 |
| | 10 | 244.70 | 290.38 | 1559.96 | 1851.18 | −643.67 | −753.74 | 1160.99 | 1387.82 | 76,623.97 | 77,678.77 |
| | 11 | 244.70 | 295.96 | 1559.96 | 1886.72 | −627.59 | −746.83 | 1177.07 | 1435.85 | 77,801.04 | 79,114.61 |
| | 12 | 244.70 | 301.64 | 1559.96 | 1922.95 | −611.51 | −739.31 | 1193.15 | 1485.27 | 78,994.20 | 80,599.89 |
| | **Total** | **2936.40** | **3267.21** | **18,719.52** | **20,828.45** | **−8399.52** | **−9233.58** | **13,256.40** | **14,862.09** | **78,994.20** | **80,599.89** |

**Table A3.** Cumulative costs for ICE and V2G solutions, occasional drivers (figures are in USD).

| CAPEX ICE | Year | Externalities ICE | Ext. ICE w. Inflation | Maintenance | Maint. w. Inflation | Fuel | Fuel w. Inflation | OPEX ICE | OPEX ICE w. Inflation | CAPEX + OPEX ICE | CAPEX + OPEX ICE w. Inflation |
|---|---|---|---|---|---|---|---|---|---|---|---|
| 35,285 | 1 | 242,30 | 242,30 | 2615.37 | 2615.37 | 285.06 | 285.06 | 3142.72 | 3142.72 | 38,427.72 | 38,427.72 |
| | 2 | 242,30 | 246,56 | 2615.37 | 2661.40 | 285.06 | 295.89 | 3142.72 | 3203.85 | 41,570.44 | 41,631.57 |
| | 3 | 242,30 | 250,90 | 2615.37 | 2708.24 | 285.06 | 307.13 | 3142.72 | 3266.27 | 44,713.17 | 44,897.84 |
| | 4 | 242,30 | 255,32 | 2615.37 | 2755.90 | 285.06 | 318.80 | 3142.72 | 3330.03 | 47,855.89 | 48,227.87 |
| | 5 | 242,30 | 259,81 | 2615.37 | 2804.41 | 285.06 | 330.92 | 3142.72 | 3395.14 | 50,998.61 | 51,623.01 |
| | 6 | 242,30 | 264,38 | 2615.37 | 2853.77 | 285.06 | 343.49 | 3142.72 | 3461.64 | 54,141.33 | 55,084.65 |
| | 7 | 242,30 | 269,04 | 2615.37 | 2903.99 | 285.06 | 356.54 | 3142.72 | 3529.57 | 57,284.06 | 58,614.22 |
| | 8 | 242,30 | 273,77 | 2615.37 | 2955.10 | 285.06 | 370.09 | 3142.72 | 3598.97 | 60,426.78 | 62,213.19 |
| | 9 | 242,30 | 278,59 | 2615.37 | 3007.11 | 285.06 | 384.16 | 3142.72 | 3669.86 | 63,569.50 | 65,883.05 |
| | 10 | 242,30 | 283,49 | 2615.37 | 3060.04 | 285.06 | 398.76 | 3142.72 | 3742.29 | 66,712.22 | 69,625.34 |
| | 11 | 242,30 | 288,48 | 2615.37 | 3113.89 | 285.06 | 413.91 | 3142.72 | 3816.29 | 69,854.94 | 73,441.62 |
| | 12 | 242,30 | 293,56 | 2615.37 | 3168.70 | 285.06 | 429.64 | 3142.72 | 3891.90 | 72,997.67 | 77,333.52 |
| | **Total** | **2907.57** | **3206.21** | **31,384.42** | **34,607.92** | **3420.67** | **4234.38** | **37,712.67** | **42,048.52** | **72,997.67** | **77,333.52** |

| CAPEX V2G | Year | Externalities V2G | Ext. V2G w. Inflation | Maintenance | Maint. w. Inflation | Electr. | Elec. w. Inflation | OPEX V2G | OPEX V2G w. Inflation | CAPEX + OPEX V2G + Battery | CAPEX + OPEX V2G w. Inflation + Battery |
|---|---|---|---|---|---|---|---|---|---|---|---|
| 44,721 | 1 | 60.84 | 60.84 | 1512.62 | 1512.62 | −1157.36 | −1157.36 | 416.11 | 416.11 | 45,137.11 | 45,137.11 |
| | 2 | 60.84 | 61.91 | 1512.62 | 1539.25 | −1141.33 | −1163.02 | 432.13 | 438.14 | 45,569.24 | 45,575.24 |
| | 3 | 60.84 | 63.00 | 1512.62 | 1566.34 | −1125.31 | −1168.07 | 448.15 | 461.26 | 46,017.39 | 46,036.51 |
| | 4 | 60.84 | 64.11 | 1512.62 | 1593.90 | −1109.29 | −1172.52 | 464.18 | 485.50 | 46,481.57 | 46,522.01 |
| | 5 | 60.84 | 65.24 | 1512.62 | 1621.96 | −1093.26 | −1176.35 | 480.20 | 510.84 | 46,961.77 | 47,032.85 |
| | 6 | 60.84 | 66.39 | 1512.62 | 1650.50 | −1077.24 | −1179.58 | 496.22 | 537.31 | 47,457.99 | 47,570.16 |
| | 7 | 60.84 | 67.55 | 1512.62 | 1679.55 | −1061.22 | −1182.20 | 512.25 | 564.91 | 47,970.24 | 48,135.07 |
| | 8 | 60.84 | 68.74 | 1512.62 | 1709.11 | −1045.19 | −1184.20 | 528.27 | 593.65 | 48,498.51 | 48,728.72 |
| | 9 | 60.84 | 69.95 | 1512.62 | 1739.19 | −1029.17 | −1185.60 | 544.29 | 623.54 | 56,048.40 | 56,357.86 |
| | 10 | 60.84 | 71.18 | 1512.62 | 1769.80 | −1013.15 | −1186.39 | 560.32 | 654.59 | 56,608.72 | 57,012.46 |
| | 11 | 60.84 | 72.44 | 1512.62 | 1800.95 | −997.12 | −1186.58 | 576.34 | 686.81 | 57,185.06 | 57,699.27 |
| | 12 | 60.84 | 73.71 | 1512.62 | 1832.65 | −981.10 | −1186.15 | 592.36 | 720.21 | 57,777.42 | 58,419.48 |
| | **Total** | **730.08** | **805.07** | **18,151.48** | **20,015.83** | **−12,830.74** | **−14,128.02** | **6050.82** | **6692.88** | **57,777.42** | **58,419.48** |

**Table A4.** Cumulative costs for battery rental and battery purchase for the V2G solution (figures are in USD).

| Year | CAPEX + Battery Rental | CAPEX + Battery Rental+ Inflation | V2G Battery Purchase | Cost Difference Rental vs. Purchase | Cost Difference Rental vs. Purchase + Inflation |
|------|------------------------|-----------------------------------|----------------------|-------------------------------------|-------------------------------------------------|
| 1 | 35,285 | 35,285 | 44,721 | −9436.00 | −9436.00 |
| 2 | 37,715.40 | 36,994.57 | 44,721 | −7005.60 | −7726.43 |
| 3 | 39,395.40 | 39,455.06 | 44,721 | −5325.60 | −5265.94 |
| 4 | 41,075.40 | 41,165.67 | 44,721 | −3645.60 | −3555.33 |
| 5 | 42,755.40 | 42,876.83 | 44,721 | −1965.60 | −1844.17 |
| 6 | 44,435.40 | 44,588.54 | 44,721 | −285.60 | −132.46 |
| 7 | 46,115.40 | 46,300.80 | 44,721 | 1394.40 | 1579.80 |
| 8 | 47,795.40 | 48,013.63 | 44,721 | 3074.40 | 3292.63 |
| 9 | 49,475.40 | 49,727.04 | 51,726.60 | −2251.20 | −1999.56 |
| 10 | 51,155.40 | 51,441.04 | 51,726.60 | −571.20 | −285.56 |
| 11 | 52,835.40 | 53,155.63 | 51,726.60 | 1108.80 | 1429.03 |
| 12 | 54,515.40 | 54,870.84 | 51,726.60 | 2788.80 | 3144.24 |

**Table A5.** Costs/profits with and without yearly degradation (professional drivers).

| Year | Bought Energy | Cost per KWh Bought | Cost per KWh + Inflation | Sold Energy | Cost per KWh Sold | Cost per KWh Sold + Inflation | Energy V2G | Energy V2G + Inflation |
|---|---|---|---|---|---|---|---|---|
| 1 | 40,000 | 0.09956 | 0.09956 | 19,695.970 | 0.131765 | 0.131765 | 1387.16 | 1387,16 |
| 2 | 39,500 | 0.09956 | 0.10145164 | 19,196.000 | 0.131765 | 0.134268535 | 1403.26 | 1429,92 |
| 3 | 39,000 | 0.09956 | 0.10334328 | 18,696.029 | 0.131765 | 0.13677207 | 1419.36 | 1473,29 |
| 4 | 38,500 | 0.09956 | 0.10523492 | 18,196.059 | 0.131765 | 0139275605 | 1435.46 | 1517,28 |
| 5 | 38,000 | 0.09956 | 0.10712656 | 17,696.088 | 0.131765 | 0.14177914 | 1451.55 | 1561,87 |
| 6 | 37,500 | 0.09956 | 0.1090182 | 17,196.118 | 0.131765 | 0.144282675 | 1467.65 | 1607,08 |
| 7 | 37,000 | 0.09956 | 0.11090984 | 16,696.147 | 0.131765 | 0.14678621 | 1483.75 | 1652,90 |
| 8 | 36,500 | 0.09956 | 0.11280148 | 16,196.177 | 0.131765 | 0.149289745 | 1499.85 | 1699,33 |
| 9 | 36,000 | 0.09956 | 0.11469312 | 15,696.206 | 0.131765 | 0.15179328 | 1515.95 | 1746,37 |
| 10 | 35,500 | 0.09956 | 0.11658476 | 15,196.236 | 0.131765 | 0.154296815 | 1532.05 | 1794,03 |
| 11 | 35,000 | 0.09956 | 0.1184764 | 14,696.265 | 0.131765 | 0.15680035 | 1548.15 | 1842,29 |
| 12 | 34,500 | 0.09956 | 0.12036804 | 14,196.295 | 0.131765 | 0.159303885 | 1564.25 | 1891,17 |
| 1 | 40,000 | 0.09956 | 0.09956 | 19,695.97 | 0.131765 | 0.131765 | 1387.16 | 1387.16 |
| 2 | 40,000 | 0.09956 | 0.10145164 | 19,695.97 | 0.131765 | 0.134268535 | 1387.16 | 1413.52 |
| 3 | 40,000 | 0.09956 | 0.10334328 | 19,695.97 | 0.131765 | 0.13677207 | 1387.16 | 1439.87 |
| 4 | 40,000 | 0.09956 | 0.10523492 | 19,695.97 | 0.131765 | 0.139275605 | 1387.16 | 1466.23 |
| 5 | 40,000 | 0.09956 | 0.10712656 | 19,695.97 | 0.131765 | 0.14177914 | 1387.16 | 1492.58 |
| 6 | 40,000 | 0.09956 | 0.1090182 | 19,695.97 | 0.131765 | 0.144282675 | 1387.16 | 1518.94 |
| 7 | 40,000 | 0.09956 | 0.11090984 | 19,695.97 | 0.131765 | 0.14678621 | 1387.16 | 1545.30 |
| 8 | 40,000 | 0.09956 | 0.11280148 | 19,695.97 | 0.131765 | 0.149289745 | 1387.16 | 1571.65 |
| 9 | 40,000 | 0.09956 | 0.11469312 | 19,695.97 | 0.131765 | 0.15179328 | 1387.16 | 1598.01 |
| 10 | 40,000 | 0.09956 | 0.11658476 | 19,695.97 | 0.131765 | 0.154296815 | 1387.16 | 1624.36 |
| 11 | 40,000 | 0.09956 | 0.1184764 | 19,695.97 | 0.131765 | 0.15680035 | 1387.16 | 1650.72 |
| 12 | 40,000 | 0.09956 | 0.12036804 | 19,695.97 | 0.131765 | 0.159303885 | 1387.16 | 1677.08 |

**Table A6.** Energy costs/profits with and without yearly degradation (frequent drivers).

| Year | Bought Energy | Cost per KWh Bought | Cost per KWh + Inflation | Sold Energy | Cost per KWh Sold | Cost per KWh Sold + Inflation | Energy V2G | Energy V2G + Inflation |
|------|---------------|---------------------|--------------------------|-------------|-------------------|-------------------------------|------------|------------------------|
| 1 | 40,000 | 0.09956 | 0.09956 | 36,207.000 | 0.131765 | 0.131765 | −788.42 | −788.42 |
| 2 | 39,500 | 0.09956 | 0.10145164 | 35,707.150 | 0.131765 | 0.134268535 | −772.33 | −787.01 |
| 3 | 39,000 | 0.09956 | 0.10334328 | 35,207.300 | 0.131765 | 0.13677207 | −756.25 | −784.99 |
| 4 | 38,500 | 0.09956 | 0.10523492 | 34,707.450 | 0.131765 | 0139275605 | −740.17 | −782.36 |
| 5 | 38,000 | 0.09956 | 0.10712656 | 34,207.600 | 0.131765 | 0.14177914 | −724.08 | −779.11 |
| 6 | 37,500 | 0.09956 | 0.1090182 | 33,707.750 | 0.131765 | 0.144282675 | −708.00 | −775.26 |
| 7 | 37,000 | 0.09956 | 0.11090984 | 33,207.900 | 0.131765 | 0.14678621 | −691.92 | −770.80 |
| 8 | 36,500 | 0.09956 | 0.11280148 | 32,708.050 | 0.131765 | 0.149289745 | −675.84 | −765.72 |
| 9 | 36,000 | 0.09956 | 0.11469312 | 32,208.200 | 0.131765 | 0.15179328 | −659.75 | −760.04 |
| 10 | 35,500 | 0.09956 | 0.11658476 | 31,708.350 | 0.131765 | 0.154296815 | −643.67 | −753.74 |
| 11 | 35,000 | 0.09956 | 0.1184764 | 31,208.500 | 0.131765 | 0.15680035 | −627.59 | −746.83 |
| 12 | 34,500 | 0.09956 | 0.12036804 | 30,708.650 | 0.131765 | 0.159303885 | −611.51 | −739.31 |
| 1 | 40,000 | 0.09956 | 0.09956 | 36,207.00 | 0.131765 | 0.131765 | −788.42 | −788.42 |
| 2 | 40,000 | 0.09956 | 0.10145164 | 36,207.00 | 0.131765 | 0.134268535 | −788.42 | −803.40 |
| 3 | 40,000 | 0.09956 | 0.10334328 | 36,207.00 | 0.131765 | 0.13677207 | −788.42 | −818.38 |
| 4 | 40,000 | 0.09956 | 0.10523492 | 36,207.00 | 0.131765 | 0.139275605 | −788.42 | −833.36 |
| 5 | 40,000 | 0.09956 | 0.10712656 | 36,207.00 | 0.131765 | 0.14177914 | −788.42 | −848.33 |
| 6 | 40,000 | 0.09956 | 0.1090182 | 36,207.00 | 0.131765 | 0.144282675 | −788.42 | −863.31 |
| 7 | 40,000 | 0.09956 | 0.11090984 | 36,207.00 | 0.131765 | 0.14678621 | −788.42 | −878.29 |
| 8 | 40,000 | 0.09956 | 0.11280148 | 36,207.00 | 0.131765 | 0.149289745 | −788.42 | −893.27 |
| 9 | 40,000 | 0.09956 | 0.11469312 | 36,207.00 | 0.131765 | 0.15179328 | −788.42 | −908.25 |
| 10 | 40,000 | 0.09956 | 0.11658476 | 36,207.00 | 0.131765 | 0.154296815 | −788.42 | −923.23 |
| 11 | 40,000 | 0.09956 | 0.1184764 | 36,207.00 | 0.131765 | 0.15680035 | −788.42 | −938.21 |
| 12 | 40,000 | 0.09956 | 0.12036804 | 36,207.00 | 0.131765 | 0.159303885 | −788.42 | −953.19 |

**Table A7.** Energy costs/profits with and without yearly degradation (occasional drivers).

| Year | Bought Energy | Cost per KWh Bought | Cost per KWh + Inflation | Sold Energy | Cost per KWh Sold | Cost per KWh Sold + Inflation | Energy V2G | Energy V2G + Inflation |
|------|---------------|---------------------|--------------------------|-------------|-------------------|-------------------------------|------------|------------------------|
| 1 | 40,000 | 0.09956 | 0.09956 | 39,007.000 | 0.131765 | 0.131765 | −1157.36 | −1157.36 |
| 2 | 39,500 | 0.09956 | 0.10145164 | 38,507.600 | 0.131765 | 0.134268535 | −1141.33 | −1163.02 |
| 3 | 39,000 | 0.09956 | 0.10334328 | 38,008.200 | 0.131765 | 0.13677207 | −1125.31 | −1168.07 |
| 4 | 38,500 | 0.09956 | 0.10523492 | 37,508.800 | 0.131765 | 0139275605 | −1109.29 | −1172.52 |
| 5 | 38,000 | 0.09956 | 0.10712656 | 37,009.400 | 0.131765 | 0.14177914 | −1093.26 | −1176.35 |
| 6 | 37,500 | 0.09956 | 0.1090182 | 36,510.000 | 0.131765 | 0.144282675 | −1077.24 | −1179.58 |
| 7 | 37,000 | 0.09956 | 0.11090984 | 36,010.600 | 0.131765 | 0.14678621 | −1061.22 | −1182.20 |
| 8 | 36,500 | 0.09956 | 0.11280148 | 35,511.200 | 0.131765 | 0.149289745 | −1045.19 | −1184.20 |
| 9 | 36,000 | 0.09956 | 0.11469312 | 35,011.800 | 0.131765 | 0.15179328 | −1029.17 | −1185.60 |
| 10 | 35,500 | 0.09956 | 0.11658476 | 34,512.400 | 0.131765 | 0.154296815 | −1013.15 | −1186.39 |
| 11 | 35,000 | 0.09956 | 0.1184764 | 34,013.000 | 0.131765 | 0.15680035 | −997.12 | −1186.58 |
| 12 | 34,500 | 0.09956 | 0.12036804 | 33,513.600 | 0.131765 | 0.159303885 | −981.10 | −1186.15 |
| 1 | 40,000 | 0.09956 | 0.09956 | 39,007.00 | 0.131765 | 0.131765 | −1157.36 | −1157.36 |
| 2 | 40,000 | 0.09956 | 0.10145164 | 39,007.00 | 0.131765 | 0.134268535 | −1157.36 | −1179.35 |
| 3 | 40,000 | 0.09956 | 0.10334328 | 39,007.00 | 0.131765 | 0.13677207 | −1157.36 | −1201.34 |
| 4 | 40,000 | 0.09956 | 0.10523492 | 39,007.00 | 0.131765 | 0.139275605 | −1157.36 | −1223.33 |
| 5 | 40,000 | 0.09956 | 0.10712656 | 39,007.00 | 0.131765 | 0.14177914 | −1157.36 | −1245.32 |
| 6 | 40,000 | 0.09956 | 0.1090182 | 39,007.00 | 0.131765 | 0.144282675 | −1157.36 | −1267.31 |
| 7 | 40,000 | 0.09956 | 0.11090984 | 39,007.00 | 0.131765 | 0.14678621 | −1157.36 | −1289.30 |
| 8 | 40,000 | 0.09956 | 0.11280148 | 39,007.00 | 0.131765 | 0.149289745 | −1157.36 | −1311.29 |
| 9 | 40,000 | 0.09956 | 0.11469312 | 39,007.00 | 0.131765 | 0.15179328 | −1157.36 | −1333.28 |
| 10 | 40,000 | 0.09956 | 0.11658476 | 39,007.00 | 0.131765 | 0.154296815 | −1157.36 | −1355.27 |
| 11 | 40,000 | 0.09956 | 0.1184764 | 39,007.00 | 0.131765 | 0.15680035 | −1157.36 | −1377.26 |
| 12 | 40,000 | 0.09956 | 0.12036804 | 39,007.00 | 0.131765 | 0.159303885 | −1157.36 | −1399.25 |

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
