# Peer review of "A Model for Cost–Benefit Analysis of Privately Owned Vehicle-to-Grid Solutions"

_energies, doi:10.3390/en13215814_

Round 1

Reviewer 1 Report

The paper defines detailed economic models for ICE, EV and EV+V2G vehicles.  ICEs versus EVs have been excessively compared previously, also V2G vs. EV has advantages (especially combined with PV generation) and drawbacks (more expensive).

The concentration of the paper on profits and savings leads in my opinion to the vague conclusions in the paper. I think that using a battery of a V2G vehicle only for tariff arbitrage is not worth the additional degradation.

The V2G main innovations are related to additional use cases such as Demand Response, flexibility and energy trading, in relation to renewable energy as mentioned above. That means that V2G is not necessarily useful for all use cases, and its deployment has to be evaluated for each scenario and user group. As an example, professional drivers use the pauses to charge the battery and less time for V2G flows, therefore the defined factors must be different.

Author Response

For the sake of readability, we send you a revised document with the editions highlighted by using red-coloured text wherever new information and references have been included.

Reviewer 1

Remark 1. The paper defines detailed economic models for ICE, EV and EV+V2G vehicles.  ICEs versus EVs have been excessively compared previously, also V2G vs. EV has advantages (especially combined with PV generation) and drawbacks (more expensive).

Answer 1: We would like to thank the reviewer for the comment that has been made and the feedback provided. We believe that the main focus of the paper is the comparison between ICE vehicles and EVs based on V2G, rather than regular EVs. Advantages and drawbacks mentioned by the reviewer have been included and highlighted.

Remark 2. The concentration of the paper on profits and savings leads in my opinion to the vague conclusions in the paper. I think that using a battery of a V2G vehicle only for tariff arbitrage is not worth the additional degradation.

Answer 2: We are grateful to the reviewer for highlighting this issue. We have added additional information to the conclusions section to better reflect the results shown in our study and have a more accurate view about the differences between ICE vehicles and EVV2Gs. We have also added some more studies (specifically, new references 24 and 25) where the economic efficiency of EVs compared to ICE vehicles is thoroughly described, especially in the long term. We believe that, considering that regular EVs might have an economic advantage in the long run over ICE vehicles, it is worth studying whether those advantages are applicable to EVV2G solutions, and if they are greater, the same or lower than by purely using EVs. Degradation, lack of degradation if the battery is rented and replacement of the automobile battery have been taken into account, too, in an effort to have figures as accurate as possible.

Remark 3. The V2G main innovations are related to additional use cases such as Demand Response, flexibility and energy trading, in relation to renewable energy as mentioned above. That means that V2G is not necessarily useful for all use cases, and its deployment has to be evaluated for each scenario and user group. As an example, professional drivers use the pauses to charge the battery and less time for V2G flows, therefore the defined factors must be different.

Answer 3: We are thankful to the reviewer for making us aware of this. Two more references regarding usage of DR and DRM have been included in order to have that perspective integrated in the manuscript. We have highlighted that EVV2G are not inevitably more economically efficient that ICE vehicles, and in fact, we reflect on how it tends to be contrary in the short term, especially if the vehicle is not used that often, hence our interest in having several groups depending on vehicle usability. Inefficiencies when recharging and trading electricity have also been included in the formulae of the mathematical model as factors that consider suboptimal situations of recharge and discharge of energy.

Reviewer 2 Report

This paper reported a cost-benefit analysis of how consumers can make use of V2G solutions, in a way that they can use their vehicle for transport purposes and obtain revenues when injecting energy into the power grid. Moreover, the paper studied how privately owned V2G can compete with ICE-based vehicles in terms of economic efficiency, putting forward some scenarios where a new mathematical model has been demonstrated, and the case study was presented. Something is new and interesting.

The paper mentioned that the potential profits that can result from applying this technology are strongly linked to battery degradation and battery costs. How about the scenario that if the battery can be rented? Can authors give some discussions?  

Author Response

For the sake of readability, we send you a revised document with the editions highlighted by using red-coloured text wherever new information and references have been included.

Reviewer: 2

Remark 1. This paper reported a cost-benefit analysis of how consumers can make use of V2G solutions, in a way that they can use their vehicle for transport purposes and obtain revenues when injecting energy into the power grid. Moreover, the paper studied how privately owned V2G can compete with ICE-based vehicles in terms of economic efficiency, putting forward some scenarios where a new mathematical model has been demonstrated, and the case study was presented. Something is new and interesting.

The paper mentioned that the potential profits that can result from applying this technology are strongly linked to battery degradation and battery costs. How about the scenario that if the battery can be rented? Can authors give some discussions? 

Answer 1: We would like to thank the reviewer for the comment that has been made and the feedback provided. We have made clearer how degradation affects the battery of a EVV2G vehicle regardless of whether it is rented or purchased. Additionally, we have included a further study where it is contemplated that the rented battery could be replaced on a yearly basis, hence effectively reducing degradation to zero in the rental scenario. Four more charts have been included to support the conclusions that have been obtained in this regard.

Reviewer 3 Report

The submission is good in conceptualization. There is both a cognitive and utilitarian motivation for the analysis conducted. The paper is interesting, but needs some work to be acceptable. I would expect deeper and more concise information:

  1. I strongly suggest that authors shall carry out more studies to compare the results from this paper to that from other similar studies. The new recent references can then be discussed in the text to improve the review quality (Introduction and Related works sections). The authors should recognize the limitations of the study. Sample reference list.
  • Weldon, P., Morrissey, P., & O’Mahony, M. (2018). Long-term cost of ownership comparative analysis between electric vehicles and internal combustion engine vehicles. Sustainable Cities and Society39, 578-591.
  • Zhang, Y., Lu, M., & Shen, S. (2020). On the values of vehicle-to-grid electricity selling in electric vehicle sharing. Manufacturing & Service Operations Management.
  • Maeng, K., Ko, S., Shin, J., & Cho, Y. (2020). How Much Electricity Sharing Will Electric Vehicle Owners Allow from Their Battery? Incorporating Vehicle-to-Grid Technology and Electricity Generation Mix. Energies13(16), 4248.
  1. Authors indicated three categories - professional drivers (that is to say, people that drive as a way to make their living), frequent drivers (people that drive on a usual basis) and occasional drivers (people that drive rarely). Please describe these categories precisely (how many kilometers per year).
  2. Authors wrote, “If this latter idea is fully taken into account, the transition from a model made up by privately owned vehicles purchased and used during a very limited amount of time every day, to one based on vehicle sharing where many different users that do not own the vehicle use it almost continuously, becomes an alternative to consider.” Please describe COVID-19 precautions for those sharing a vehicle. The authors should explain further research possibilities.
  3. Poor edition, not following the manuscript preparation instructions (references). Errors in the text (“Table 43”, etc.)

Author Response

For the sake of readability, we send you a revised document with the editions highlighted by using red-coloured text wherever new information and references have been included.

Reviewer: 3

Remark 1. The submission is good in conceptualization. There is both a cognitive and utilitarian motivation for the analysis conducted. The paper is interesting, but needs some work to be acceptable. I would expect deeper and more concise information:

  1. I strongly suggest that authors shall carry out more studies to compare the results from this paper to that from other similar studies. The new recent references can then be discussed in the text to improve the review quality (Introduction and Related works sections). The authors should recognize the limitations of the study. Sample reference list.
  • Weldon, P., Morrissey, P., & O’Mahony, M. (2018). Long-term cost of ownership comparative analysis between electric vehicles and internal combustion engine vehicles. Sustainable Cities and Society, 39, 578-591.
  • Zhang, Y., Lu, M., & Shen, S. (2020). On the values of vehicle-to-grid electricity selling in electric vehicle sharing. Manufacturing & Service Operations Management.
  • Maeng, K., Ko, S., Shin, J., & Cho, Y. (2020). How Much Electricity Sharing Will Electric Vehicle Owners Allow from Their Battery? Incorporating Vehicle-to-Grid Technology and Electricity Generation Mix. Energies, 13(16), 4248.

Answer 1: We would like to thank the reviewer for the comment that has been made and the effort put into the review of the paper. The studies that have been mentioned by the reviewer, along with several more references related to other requests have been included in the state of the art so it is more complete. The references have been studied; their contributions and how they compare with our manuscript has been included.

Remark 2. Authors indicated three categories - professional drivers (that is to say, people that drive as a way to make their living), frequent drivers (people that drive on a usual basis) and occasional drivers (people that drive rarely). Please describe these categories precisely (how many kilometers per year).

Answer 2: We are grateful to the reviewer for highlighting this issue. We have clarified how we have conceived the three different categories in terms of yearly mileage and gas consumption. We have explicitly mentioned what references we are making use of (47, 55, 63, 64) right after Table 4 and how we have used their data in order to infer the three categories of the manuscript.

Remark 3. Authors wrote, “If this latter idea is fully taken into account, the transition from a model made up by privately owned vehicles purchased and used during a very limited amount of time every day, to one based on vehicle sharing where many different users that do not own the vehicle use it almost continuously, becomes an alternative to consider.” Please describe COVID-19 precautions for those sharing a vehicle. The authors should explain further research possibilities.

Answer 3: We are thankful to the reviewer for this request, as we had not included any content related to COVID-19. Conclusions have been further enhanced in order to include this kind of information and what kind of precautions can be taken. Two references related to possible future solutions for this challenge have been included as well.

Remark 4. Poor edition, not following the manuscript preparation instructions (references). Errors in the text (“Table 43”, etc.)

Answer 4: We thank the reviewer for making us aware of this problem. All the references have been changed to match the pattern that is used in Energies. Text and numbers in figures and tables has been reviewed.

Round 2

Reviewer 1 Report

In authors' Answer 2 they write: " ... it is worth studying whether those advantages are applicable to EVV2G solutions, and if they are greater, the same or lower than by purely using EVs". Comparing the total cost for pure EVs against EVV2G would indeed improve the overall picture. If the costs elements are not available for the calculation, there are studies that can have done this comparison and can be cited instead.

Another remark to the Tables 11 refers to the relationship between the yearly bought energy e.g 40,000kWh and the sold one in V2G operation, 19,000kWh up to 39,000kWh (for occasional drivers). The amount of sold energy corresponds to deep battery cycles that accelerate its deterioration. Please comment on how the numbers have been selected.

Typo in line 887: you meant probably renting instead purchasing

Author Response

For the sake of readability, we send you a revised document with the editions highlighted by using red-coloured text wherever new information and references have been included.

Reviewer 1

Remark 1. In authors' Answer 2 they write: " ... it is worth studying whether those advantages are applicable to EVV2G solutions, and if they are greater, the same or lower than by purely using EVs". Comparing the total cost for pure EVs against EVV2G would indeed improve the overall picture. If the costs elements are not available for the calculation, there are studies that can have done this comparison and can be cited instead.

Answer 1. We thank the reviewer for this comment, as it has helped us enhancing the content of the paper. We have added three additional references where comparisons between economic performance of EVs and V2G solutions are compared. Specifically, reference [32] provides four different schemes with the two ones involving V2G being mentioned as a way to profit from electricity arbitrage. References [31] and [33] refer to several aspects of V2G charging for EVs, such as scheduling and EV-to-V2G/EV communications to provide their own capability to the power grid.

Remark 2. Another remark to the Tables 11 refers to the relationship between the yearly bought energy e.g 40,000kWh and the sold one in V2G operation, 19,000kWh up to 39,000kWh (for occasional drivers). The amount of sold energy corresponds to deep battery cycles that accelerate its deterioration. Please comment on how the numbers have been selected.

Answer 2. We are grateful for the reviewer for letting us know about this. We have added an explanation in Section 5.1 about how the calculations for energy expenses have been done, which are extensible to all the 3 scenarios that have been put forward. It has been considered that the battery that is used by an average EV is 40KWh. Taking into account a) the mileage that an EVV2G is driven every year and b) the battery degradation that takes place after several years, a diminishing amount of energy available to trade has been considered for each situation, which will be more or less abundant depending on how. Calculations on the quantity of energy used for every have been supported with reference [61] in section 5.1. We agree on the fact that this way of using the battery in a V2G accelerates its degradation; that is why significant degradation coefficients have been considered for the battery along with a battery replacement after eight years of using the V2G solution, which are not commonly considered for regular EVs

Remark 3. Typo in line 887: you meant probably renting instead purchasing.

Answer 3. We thank the reviewer for highlighting this issue. Indeed, we meant renting. It has been changed to that.

Reviewer 3 Report

The authors have carefully reviewed all suggestions, and changes have been made and marked to the last manuscript where appropriate. Explanation and corresponding revisions prove sufficient to the mentioned concerns and render the manuscript acceptable for publication.

Author Response

For the sake of readability, we send you a revised document with the editions highlighted by using red-coloured text wherever new information and references have been included.

Reviewer 3

Remark 1. The authors have carefully reviewed all suggestions, and changes have been made and marked to the last manuscript where appropriate. Explanation and corresponding revisions prove sufficient to the mentioned concerns and render the manuscript acceptable for publication.

Answer 1. We thank the reviewer for this comment. We have tried our best to improve the manuscript.
